# Deep-learning structure elucidation from single-mutant deep mutational scanning

Zachary C. Drake[1], Elijah H. Day[1], Paul D. Toth[2] & Steffen Lindert [1] ✉

Deep learning has revolutionized the field of protein structure prediction. AlphaFold2, a deep neural network, vastly outperformed previous algorithms to provide near atomic-level accuracy when predicting protein structures. Despite its success, there still are limitations which prevent accurate predictions for numerous protein systems. Here we show that sparse residue burial restraints from deep mutational scanning (DMS) can refine AlphaFold2 to significantly enhance results. Burial information extracted from DMS is used to explicitly guide residue placement during structure generation. DMS-Fold was validated on both simulated and experimental single-mutant DMS, with DMS-Fold outperforming AlphaFold2 for 88% of protein targets and with 252 proteins having an improvement greater than 0.1 in TM-Score. DMS-Fold is free and publicly available: [https://github.com/LindertLab/DMS-Fold].

Deep learning has led to groundbreaking advancements in the field of protein structure prediction. The emergence of AlphaFold2[1], which harnesses evolutionary patterns present in protein sequences, demonstrated the feasibility of predicting protein structures with unprecedented accuracy at a large scale[2]. Its success has not only revolutionized the field but has also inspired other deep-learning-based structure predictors such as RosettaFold[3], ESMFold[4], OmegaFold[5], and more[6]. Despite its remarkable success, AlphaFold2 still has several limitations and challenges such as handling dynamic proteins with multiple conformations, predicting mutational effects, orphan proteins or intrinsically disordered proteins[7–9]. The field of protein structure prediction is continuously improving as deep learning continues to evolve and new computational methods are devised. In addition to algorithmic refinement, a potentially more promising avenue for enhancing performance lies in the integration of sparse data with deep learning methods.

Sparse data can reveal key protein structural insights which can be utilized to improve computational tools for protein structure prediction. Previous work demonstrated that incorporating sparse data from techniques such as structural mass spectrometry and nuclear magnetic resonance into conventional physics/knowledge-based algorithms, such as Rosetta, led to more accurate predictions[10–27]. When combined with deep learning, sparse data can be used to develop models which outperform AlphaFold2, such as using experimental contacts from photo-crosslinking mass spectrometry[28,29] or density maps from cryo-electron microscopy[30].

Deep mutational scanning (DMS) is a versatile, high throughput technique that systematically maps genetic variations to phenotypes[31]. Different types of assays can be employed to evaluate mutational sensitivities and effects on thermodynamic stability or enzymatic activity[32]. DMS experiments produce large, information-rich datasets which are ideal for developing deep learning models. Multiple models have been developed and trained to predict protein sequence-function relationships[33–35]. Conventional, non-machine-learning protocols have made significant advances in inferring structural information of proteins from DMS and experimental evolution coupling data[36–41]. However, the actual three-dimensional protein structure prediction from DMS data has been underdeveloped. We propose that leveraging structural information from DMS in conjunction with deep learning can accurately predict protein structure and overcome the remaining limitations of AlphaFold.

In this work, we present DMS-Fold: a deep neural network trained on residue burial restraints derived from single-mutant DMS. DMS-Fold consolidates structural information extracted from DMS within an AlphaFold2 framework. We propose a method of predicting residue burial information from DMS by analyzing protein thermodynamic folding stabilities from a mega-scale set of proteins[42]. We demonstrate that this information can be used to explicitly guide residue placement with DMS-Fold, improving AlphaFold2 predictions of 89% and 85% of protein targets with simulated and experimental mutagenic data, respectively.

[1]Department of Chemistry and Biochemistry, University of California, Los Angeles, Los Angeles, CA, USA. [2]Department of Chemistry and Biochemistry, Ohio State University, Columbus, OH, USA. ✉e-mail: lindert@ucla.edu

## Results and Discussion

### Extracting burial information from deep mutational scanning and integration into DMS-fold

Protein tertiary structures typically exhibit a high concentration of hydrophobic residues in the core and exposed hydrophilic residues at the surface[43]. Point mutations that convert hydrophobic core residues into polar/charged residues are likely to cause significant disruptions in dynamics and stability, potentially leading to misfolding[44]. Thus, our hypothesis was that we could infer the distance of a residue from the surface (i.e., surface distance) of a protein by assessing the detrimental effects of different mutational types on protein folding stability in a large dataset of deep mutational scanning (DMS) data for numerous proteins with known native structures. Tsuboyama et al.[42] published a mega-scale single-mutant deep mutational scanning (DMS) dataset evaluating the effects of mutations on folding stabilities across 331 natural and 148 de novo designed proteins. Approximately 776,000 high-quality folding stabilities were measured using cDNA display proteolysis, a high-throughput stability assay.

To investigate for which mutational type locations within the protein most strongly correlated with folding stabilities, we compared residue surface distances of 175 proteins in the mega-scale set with changes in protein thermodynamic stabilities (ΔΔGs) of individual point mutations (Fig. 1). Specifically, we quantified residue surface distances using two different solvent exposure metrics: neighbor count and atomic depth (as detailed in the methods 'Burial score and burial metrics' section). We calculated the correlations, represented as coefficients of determination ($R^2$), between mutational stability and both solubility metrics independently (Fig. 1a, b). Notably, we observed variations in correlations for specific mutational types (Fig. 1c), with one of the metrics showing stronger associations. To enhance overall correlations, we introduced a weighted average combining both metrics, which we termed 'burial extent' (Fig. 1d).

In general, mutations from small nonpolar residues to charged/polar residues were observed to have the greatest correlation between a residue's burial extent and mutational stability. As indicated in Supplementary Table 1, mutating aliphatic residues (A,V, and I) to charged/polar residues (N, K, Q, E, H, D, S, R, and T) were the most likely to destabilize protein folding for deeper residues. Interestingly, mutating buried tryptophan residues to glycine had the seventh highest correlation, which could be attributed to dramatic size changes or possible disruption of secondary structure. We compared these mutational type correlations to scores from the BLOSUM62 substitution matrix by calculating Z-scores (Supplementary Fig. 1), excluding cysteine residues which did not have considerable mutational coverage[45]. High or low Z-Scores indicate our analysis predicts particular mutations to be more detrimental or less detrimental, respectively, as compared to

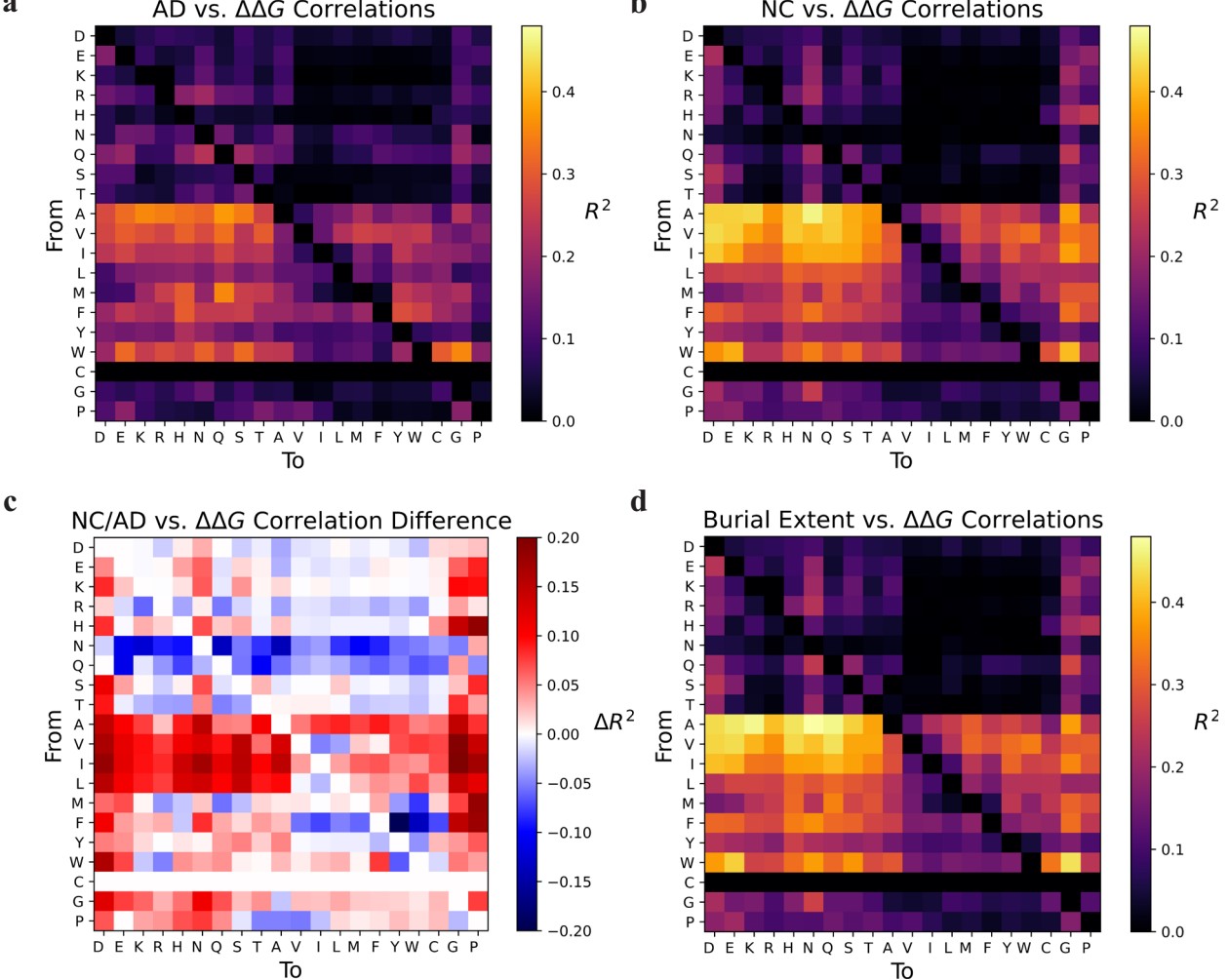

**Fig. 1 | Heat maps depicting the correlation between changes in protein thermodynamic stabilities (ΔΔG) and solubility metrics, atomic depth (AD) and neighbor count (NC), for individual mutational types across the mega-scale set. a** Comparing ΔΔGs to native residue atomic depth. **b** Comparing ΔΔGs to native neighbor count. **c** Differences in correlation coefficients between atomic depth and neighbor count. **d** Comparing ΔΔGs to burial extent (defined as weighted average of both atomic depth and neighbor count).

BLOSUM while zero values indicate agreement. Notable discrepancies between our method and BLOSUM involved mutations which would not typically be considered as structurally disruptive based on our analysis that focuses on burial extent. For example, large Z-scores were observed for polar/charged residues substituted for nonpolar/non-charged residues, or when small aliphatic residues were replaced with other small aliphatic residues. These mutational types may have low BLOSUM scores due to their functional importance but are not predicted to be structurally disruptive through our analysis. Examples for such mutations include residues in a binding site or on the protein surface.

## DMS-Fold: combining DMS with deep learning structure prediction

OpenFold[46], a trainable and identical reproduction of AlphaFold2[1], has been shown to be a successful approach for integrating sparse data into neural-networks[28]. In this work, we introduce DMS-Fold, a version of OpenFold which utilizes burial information obtained from DMS experiments. Although AlphaFold2 captures underlying principles related to residue buriedness and its evolutionary significance[47,48], DMS data can provide a more effective and detailed understanding through empirical and context-specific insights[49,50]. The OpenFold/AlphaFold2 framework processes a given protein sequence along Multiple Sequence Alignments (MSA) of homologous proteins, translating these inputs into two distinct formats. One is the MSA representation, which encodes the target sequence and the aligned sequences. The other is the pair representation, whose elements contain residue pair information. The main block of the network, called the Evoformer, updates these representations by leveraging co-evolutionary information derived from MSAs. Subsequently, the updated representations are then transformed by the structure module into 3D coordinates of a protein structure. Although residue burial information from single-mutant DMS does not directly reflect residue contacts, it indirectly encodes residue densities. Buried residues are likely to be in close proximity to other buried residues. We hypothesized that by embedding predicted residue surface distances into the pair representation, the MSA transformer could be biased to correctly place these as core/surface residues during retrieval of co-evolutionary information.

To predict and quantify residue surface distance from DMS data, we developed a metric termed 'burial score' which averages ΔΔGs of different mutations of a specific residue weighted by mutational type correlations calculated from the mega-scale set (Fig. 2a). A low burial score indicates structurally disruptive mutations and predicts the target residue to likely be submerged within the core of the protein, while a high burial score would indicate the opposite. DMS data was added as an additional feature to the network as well as a recycle embedder which embedded these encoded burial scores along the diagonal of the pair representation during initialization prior to Evoformer processing (Fig. 2b). We used two types of burial scores in this study: "encoded burial scores", where higher values indicate residues predicted to be buried, and "unencoded burial scores" which represent weighted and averaged ΔΔGs, meaning that lower values indicate residues predicted to be buried. By embedding along the diagonal, specific pair information was not distorted.

A curated training set of 15,392 structurally diverse proteins was obtained using the PISCES sequence culling webserver[51]. ΔΔGs of these proteins were simulated using ThermoMPNN[52], a graph neural network trained on the mega-scale set to predict thermodynamic stabilities from a PDB structure. DMS-Fold was initialized with AlphaFold2's weights and trained for 5 epochs, following OpenFold's training regimen.

## Using burial information from simulated protein folding stabilities improved 89% of predictions when using DMS-fold

A set of 710 protein targets from the CASP14[53] and CAMEO[54] sets were used to evaluate the performance of DMS-Fold. Folding stabilities of point mutations were simulated for these proteins using ThermoMPNN[52]. TM-Scores averaged across 25 predictions from DMS-Fold were compared to AlphaFold2 using AlphaFold2's 'model_5_ptm' weights (Fig. 3a, b). DMS-Fold outperformed AlphaFold2 for 631 targets (with MSA subsampling using a size-dependent $N_{eff}$ for both networks) with an average TM-Score improvement of 0.08. Higher TM-Score improvements over AlphaFold2 predictions generally corresponded to systems with greater network confidence (Fig. 3a). When comparing distributions of binned predictions across all levels of AlphaFold2 accuracy, DMS-Fold was consistently more accurate (Fig. 3b).

Predictions for both networks were done at various $N_{eff}$ values to simulate challenging targets and to gauge the relationship between the reliance on MSAs and the influence from DMS burial information (Fig. 3c). At lower $N_{eff}$ values, inclusion of DMS data lead to dramatically improved predictions. The number of predictions not in the correct fold (TM-Score <0.5) for AlphaFold2 and DMS-Fold were 578 and 225, when using a $N_{eff}$ of 1.0, and 149 and 82 when using a $N_{eff}$ of 5.0. At higher $N_{eff}$ values, a deeper MSA pool was used, resulting in an increased accuracy for AlphaFold2. However, even when using full MSAs (no MSA-subsampling), influences from burial information could be seen in the tighter distribution of predictions for DMS-Fold, with a distribution interquartile range (IQR) of 0.08 for DMS-Fold and 0.10 for AlphaFold2. When comparing predictions from DMS-Fold with a size-dependent $N_{eff}$ and AlphaFold2 with no MSA subsampling, DMS-Fold outperformed AlphaFold2 for 267 systems, with 49 DMS-Fold predictions having TM-Score improvements ≥ 0.1. Figure 3d compares the top five predictions with the greatest improvement in TM-Score with DMS-Fold (size-dependent $N_{eff}$) with respect to AlphaFold2 (no MSA-subsampling) aligned to the native structure.

Network confidence was examined when comparing DMS-Fold/AlphaFold2 predictions. Changes in the AlphaFold2 per-residue confidence metric, predicted local distance different test (pLDDT), were compared to changes in TM-Score and differences from native structure solubility metrics (Fig. 3e). Predictions which improved in accuracy from the inclusion of DMS burial information also showed greater network confidence through higher pLDDTs. Predicted DMS-Fold structures which were more confident also had more nativelike burial metrics (color mapping in Fig. 3e).

## Using burial information from experimental protein folding stabilities improved predictions for mega-scale targets when using DMS-fold

To demonstrate how the network handles experimental DMS data, predictions of the 175 proteins in the mega-scale set were generated with both DMS-Fold and AlphaFold2. DMS-Fold predictions were generated through the application of dataset splitting (see section 'Avoiding bias in mega-scale benchmarking with dataset splitting' in methods) to minimize structural homology between the sets and to ensure each prediction avoids any bias or data leakage. Mutational type correlations were calculated for two separate subsets of the mega-scale set (Supplementary Fig. 2 and Supplementary Fig. 3) and yielded nearly identical trends as those in Fig. 1. These correlations were then used to train two independent versions of DMS-Fold so that predictions of each the mega-scale proteins avoided any potential bias. DMS-Fold predictions employing ΔΔGs taken from Tsobutoma et al.[42] were compared to AlphaFold2 predictions (Fig. 4a, b). When comparing TM-Scores averaged across 25 random seeds, DMS-Fold outperformed AlphaFold2 for 149 targets (with MSA subsampling and a size-dependent $N_{eff}$) with an average TM-Score improvement of 0.17. Similar to the CASP14/CAMEO sets, proteins with larger TM-Score improvements displayed greater network confidence (Fig. 4a), and DMS-Fold predictions were generally more accurate across all levels of AlphaFold2 accuracy (Fig. 4b). Supplementary Fig. 4 shows the comparison of AlphaFold2 to DMS-Fold trained with burial scores derived

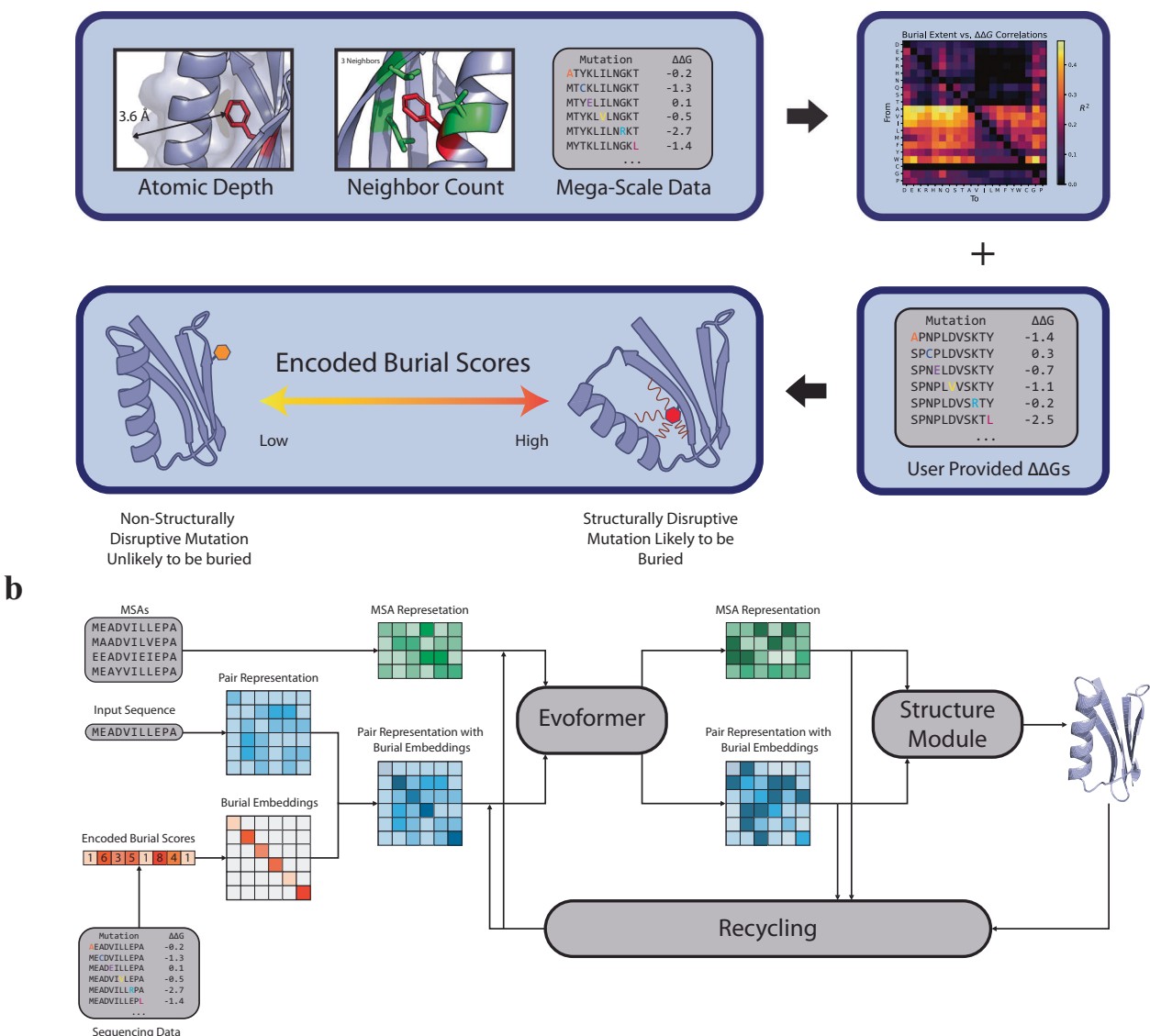

**Fig. 2 | DMS-fold overview. a** Atomic depth, neighbor count, and mutation ΔΔGs are used to identify mutational types likely to be destabilizing for buried residues. These mutational types are used to calculate burial scores of residues from given mutational stabilities. **b** DMS-Fold network architecture based on the original

from THPLM, which resulted in 115 proteins improving with an average TM-Score increase of only 0.08. The poorer performance of this DMS-Fold model is most likely due to the difficulty of accurately predicting stabilities from only sequence information (as demonstrated in Supplementary Fig. 5). It was observed that residues with larger encoded burial scores generally were more likely to be modeled more accurately. This can be seen in Supplementary Fig. 6a, b, where residues with low encoded burial scores of 0 and 1 had greater variability of α-carbon (CA) per-residue root-mean-squared deviations (RMSD) with maximum values of 20.0 Å and 15.7 Å, respectively while higher encoded burial scores of 8 and 9 resulted in maximum RMSDs of 3.7 Å and 2.2 Å. This underscores that the information encoded in the burial score (i.e., the fact that a residue is located in a buried location in the

OpenFold architecture. Residue burial information derived from deep mutational scanning data (**a**) is encoded as burial scores. These are then embedded into the pair representation along the diagonal. The pair representation, coupled with the MSA representation, is initialized before being processed by the Evoformer.

core of the protein) allowed us to more accurately restrain those residues to their correct positions. Figure 4d compares AlphaFold2/DMS-Fold predictions for five proteins with the largest TM-Score improvement.

As with the CASP14/CAMEO set, predictions for both networks were done at various $N_{eff}$ values to assess the impact of burial information from experimental DMS on the modeling results (Fig. 4c). A narrower $N_{eff}$ range was used to account for shorter sequences in the mega-scale set. At lower $N_{eff}$ values, greater DMS influence from experimental data could be seen by comparing the TM-Score distributions of two networks. We also observed that manually masking MSAs of residues based on burial score had larger effects on AlphaFold2 then DMS-Fold (Supplementary Fig. 7). The number of

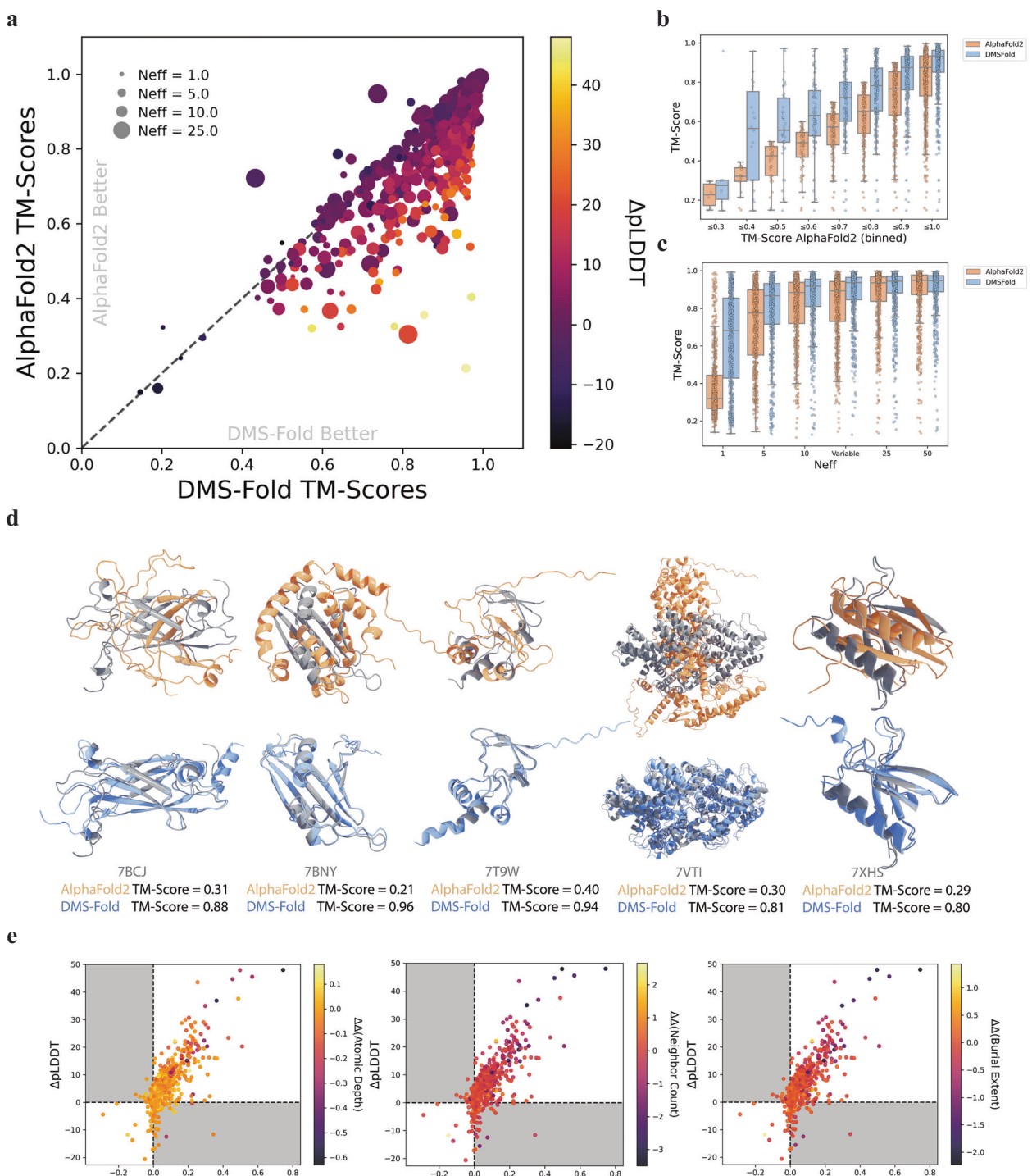

**Fig. 3 | Performance of DMS-Fold on the 710 CASP14/CAMEO proteins with simulated changes in protein thermodynamic stabilities (ΔΔGs). a** Template modeling score (TM-Score) comparison of predictions from DMS-Fold and AlphaFold2 ($N = 25$) using a size-dependent number of nonredundant sequences ($N_{eff}$). Size of each marker represents the $N_{eff}$ used for MSA subsampling. Color represents the change in network confidence, pLDDT, between DMS-Fold and AlphaFold2. **b** TM-Score distributions of both networks binned to TM-Scores of AlphaFold2 predictions ($N = 25$). **c** TM-Score distributions of predictions from both DMS-Fold and AlphaFold2 ($N = 1$) using different uniform $N_{eff}$ values. **d** Five

predicted structures (aligned to native structure (grey)) where DMS-Fold with a size-dependent $N_{eff}$ (blue) had a TM-Score improvement > 0.5 compared to AlphaFold2 with no MSA-subsampling (orange). **e** Comparison of changes in pLDDTs and TM-Scores between predictions with DMS-Fold and AlphaFold2. Color represents the change in the difference of solubility metrics for the DMS-Fold structure and the native structure with the AlphaFold2 structure and the native structure. Points in panels a and b show the mean. In all box plots, the line shows the median and the whiskers represent the 1.5x interquartile range.

predictions not in the correct fold (TM-Score <0.5) for AlphaFold2 and DMS-Fold were 92 and 11, respectively, when only using a $N_{eff}$ of 1.0, and 40 and 6 when using a $N_{eff}$ of 2.0. Higher $N_{eff}$ values resulted in increased accuracy for AlphaFold2. When using full MSAs (no MSA-

subsampling), the AlphaFold2 TM-Score distribution was statistically identical to the DMS-Fold distribution, with a distribution IQR of 0.153 and 0.156 for DMS-Fold and AlphaFold2, respectively, and a p-value of 0.50 indicating no significant difference according to the Mann-

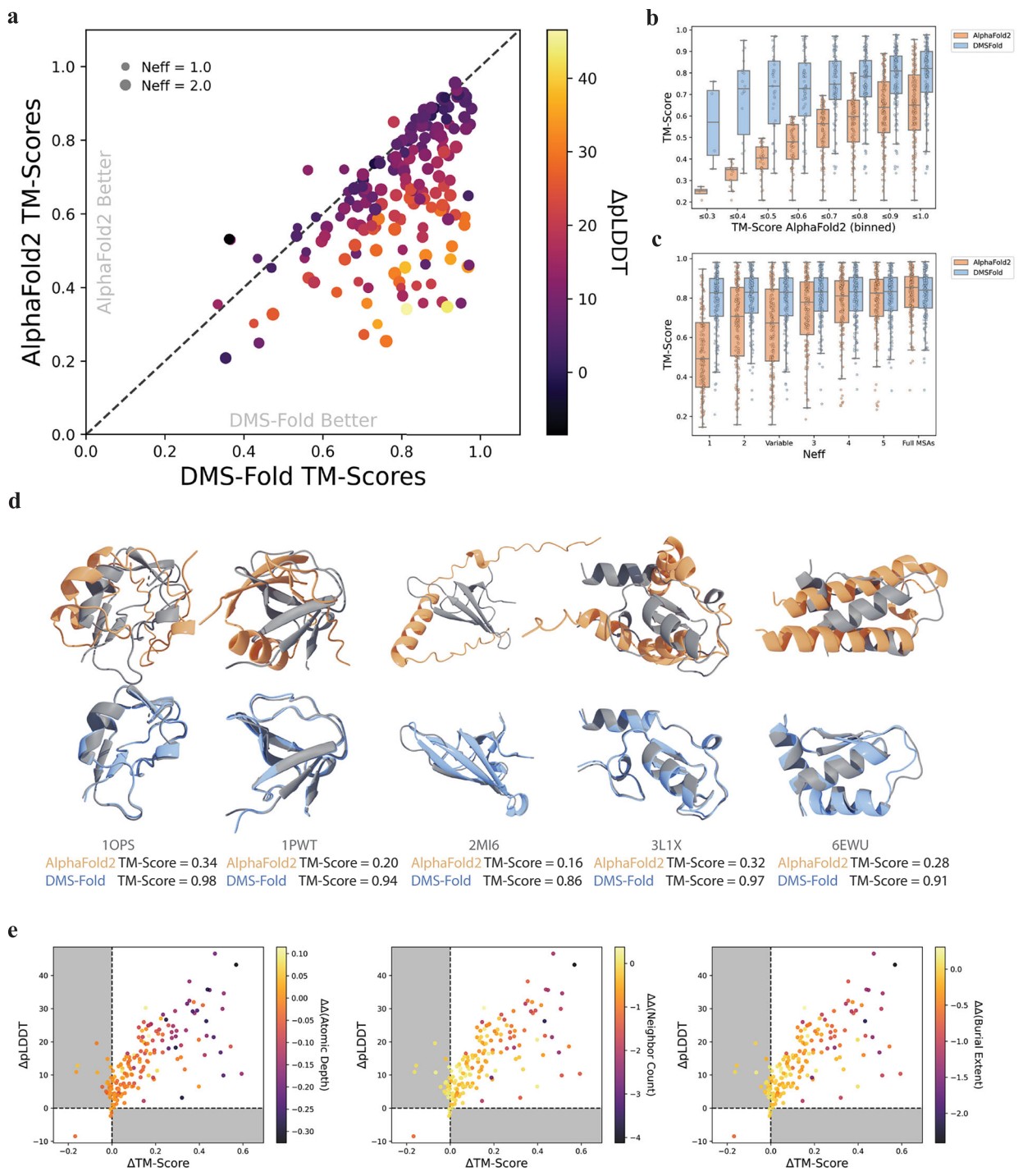

**Fig. 4 | Performance of DMS-Fold on the 175 Mega-scale proteins using experimental changes in protein thermodynamic stabilities ($\Delta\Delta G$s). a** Template modeling score (TM-Score) comparison of predictions from DMS-Fold and AlphaFold2 ($N = 25$) using a size-dependent number of nonredundant sequences ($N_{eff}$). Size of each marker represents the $N_{eff}$ used for MSA subsampling. Color represents the change in network confidence, pLDDT between DMS-Fold and AlphaFold2. **b** TM-Score distributions of both networks binned to TM-Scores of AlphaFold2 predictions ($N = 25$). **c** TM-Score distributions of predictions from both DMS-Fold and AlphaFold2 using different uniform $N_{eff}$ values. **d** Top five predicted structures from AlphaFold2 with a size-dependent $N_{eff}$ (orange) and DMS-Fold ($N = 1$) with a size-dependent $N_{eff}$ (blue) aligned to their native structure (grey). **e** Comparison of changes in pLDDTs and TM-Scores between predictions with DMS-Fold and AlphaFold2. Color represents the change in the difference of solubility metrics for the DMS-Fold structure and the native structure with the AlphaFold2 structure and the native structure. Points in panels (**a**, **b**) show the mean. In all box plots, the line shows the median and the whiskers represent the 1.5x interquartile range.

Whitney U test ($p$-value $> 0.05$). However, DMS-Fold with a size-dependent $N_{eff}$ still outperformed AlphaFold2 without MSA-subsampling for 57 proteins with 5 of them having TM-Score improvements $\geq 0.1$.

Network confidence was also examined when considering DMS-Fold predictions using experimental $\Delta\Delta G$s. Changes in the pLDDT were compared to changes in TM-Score and differences from native structure solubility metrics (Fig. 4e). Similar to the results observed for the

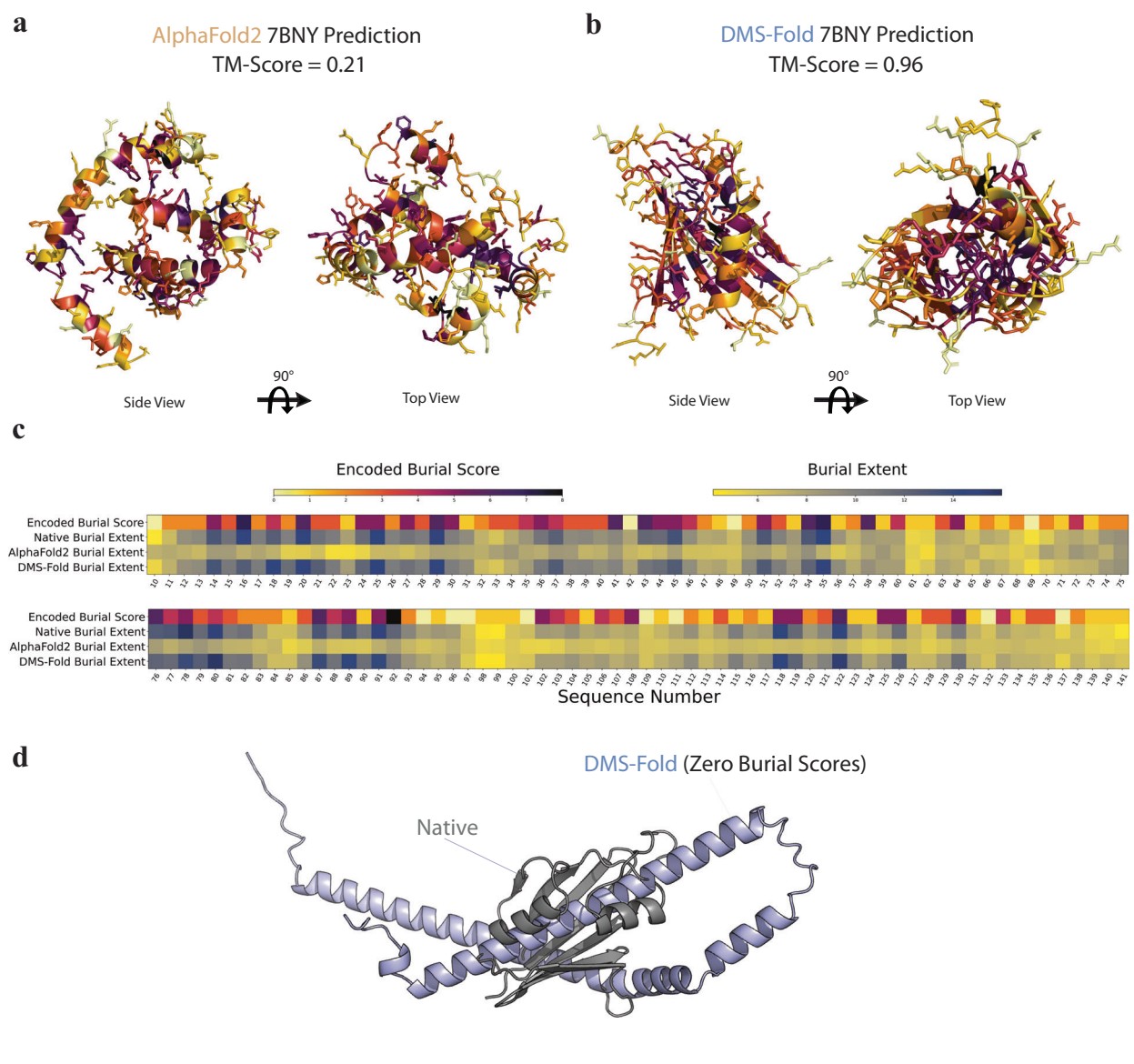

**Fig. 5 | Burial scores explicitly guide DMS-Fold inference. a** AlphaFold2 prediction of protein 2 A (PDB ID: 7BNY) with residues colored by encoded burial score (legend in panel **c**). **b** DMS-Fold prediction of protein 2 A with residues colored by encoded burial score. **c** Per-residue comparisons of predicted encoded burial scores and burial extents of the native, AlphaFold2, and DMS-Fold structures. **d** DMS-Fold prediction of protein 2 A using false encoded burial scores of zero for all residues (blue) compared to native structure (grey).

CASP14/CAMEO set, DMS-Fold predictions improved in accuracy from the inclusion of experimental DMS burial information while simultaneously exhibiting greater network confidence. Although the mega-scale set was split into two structurally distinct groups, we investigated potential data leakage from including the mega-scale proteins at all in the DMS-Fold training set. To address this, we retrained DMS-Fold excluding all mega-scale proteins from the training data. As shown in Supplementary Fig. 8, predictions on the mega-scale set using these models were consistent with the results presented in Fig. 4, leading us to conclude that no data leakage from including the mega-scale proteins was present.

**Burial Scores Explicitly Guide Residue Placement in DMS-Fold Inference**

In a case study to investigate the impact of burial scores on DMS-Fold predictions, we examined the predicted structures for the 2 A protein from encephalomyocarditis virus (PDB ID: 7BNY)[55] using AlphaFold2 (no MSA-subsampling) and DMS-Fold (size-dependent $N_{eff}$), respectively. AlphaFold2 predicted a diffuse structure comprised of multiple alpha-helices (Fig. 5a). The structure had a low contact order of 3.6, leaving hydrophobic residues exposed[56]. In contrast, the DMS-Fold prediction positioned and oriented residues with high encoded burial scores towards the core of the protein (Fig. 5b). The DMS-Fold structure had a high contact order of 16.0, and properly buried hydrophobic residues. The additional information about residue burial allowed DMS-Fold to correctly predict key structural features, such as 2As RNA-binding fold, comprised of seven antiparallel beta strands and two alpha-helices.

In Fig. 5c, per-residue encoded burial scores and burial extents are compared between the native crystal structure, the DMS-Fold prediction, and the AlphaFold2 prediction. Residues with high predicted encoded burial scores were generally observed to be buried in the

native structure, as seen from agreement with burial extents. Alpha-Fold2's diffuse structure resulted in somewhat uniform low burial extents, contrasting with DMS-Fold's nativelike structure which showed strong agreement with both native burial extents and predicted encoded burial scores. Furthermore, we tested burial score influence on DMS-Fold by assigning all residues of protein 2 A false encoded burial scores equal to zero, to see if this would be reflected in DMS-Fold inference. DMS-Fold interpreted these low encoded burial scores as more exposed residues, predicting a structure of primarily alpha-helical secondary structure content (Fig. 5d). This demonstrates that DMS-Fold has learned to interpret the burial scores as explicit exposure constraints.

## Discussion

Similar to previous work that showed correlations between deep mutational scanning (DMS) fitness/tolerance and solvent accessible surface area[57,58], we demonstrated that residue burial information can be extracted from single-mutant deep mutational scanning. Solubility metrics were correlated to changes in thermodynamic folding stabilities ($\Delta\Delta$Gs) of proteins in the mega-scale set[42]. We found that mutations which introduce charge (or at least polarity) are most likely to be deleterious for residues farther from the protein surface. We developed a metric termed 'burial score' which predicts residue location relative to the surface by averaging $\Delta\Delta$Gs weighted by correlations from the mega-scale set.

We subsequently presented DMS-Fold, a deep neural network which utilizes burial information from DMS as experimental restraints within AlphaFold to enhance predictions. Burial scores calculated from mutagenic data were embedded within the pair representation through a recycling embedder, biasing retrieval of co-evolutionary information. DMS-Fold was trained on a curated set of 15,392 proteins with simulated $\Delta\Delta$Gs generated using ThermoMPNN. During inference, burial scores were used to explicitly guide residue position with respect to the interior of a protein.

DMS-Fold was validated on both simulated $\Delta\Delta$Gs for various CASP14/CAMEO targets and experimental $\Delta\Delta$Gs for proteins in the mega-scale set. When using DMS data, major improvements were seen for both sets when comparing predictions from DMS-Fold and AlphaFold2, with DMS-Fold outperforming AlphaFold2 for 89% of CASP14/CAMEO targets and 85% of mega-scale targets. 23% and 51% of targets had TM-score improvements greater than 0.1 for the CASP14/CAMEO and mega-scale sets, respectively. Greater improvements coincided with higher network confidence. Different levels of MSA subsampling were explored to evaluate the relationship between DMS and co-evolutionary information. Shallower MSAs resulted in heavier reliance on burial information. By applying dataset splitting to the training of both ThermoMPNN and DMS-Fold, we ensure that each prediction for the mega-scale set remained independent. The significant improvement observed for the mega-scale set, combined with our use of experimentally derived burial scores, suggests no evident source of data leakage. We observed improvements with an alternative model, THPLM, which relies solely on protein sequences to simulate point mutation stabilities. Although DMS-Fold trained with THPLM stabilities performed worse than the ThermoMPNN-based model, a notable number of the mega-scale targets showed improvement, indicating that DMS-Fold may be useful when paired with a sequence-based predictor of point mutation stabilities. Other types of DMS data (e.g., functional or enrichment scores) may potentially be compatible with DMS-Fold, assuming they correlate with $\Delta\Delta$Gs and have high mutational coverage (mega-scale proteins had 93.4% mutational coverage). Future work is required to validate this conclusion.

Our work shows that beyond studying protein function, stability, roles of amino acids, and mutational effects, single-mutant deep mutational scanning (DMS) can be incredibly beneficial in protein structure prediction. DMS-Fold demonstrated that additional experimental restraints from DMS can be used to refine AlphaFold2's network parameters to enhance de novo protein structure prediction by biasing co-evolutionary retrieval and explicitly guiding residue placement. Massively parallel experiments, such as DMS, are increasingly becoming more efficient and accessible. As massive amounts of experimental data become more available, hybrid protein structure methods will continue to become more established and more efficient to develop. We demonstrate that limitations of AlphaFold2 and other deep learning models can be overcome with structural insights from sparse data. In the near future, accurate predictions from hybrid deep learning models may be feasible for nearly any potential protein system. Although DMS-Fold currently only supports the use of mutagenic data for predicting structures of wild type sequences, future work could focus on expanding the network to predict the structural effects of deleterious mutations or even identify lethal mutations, similar to AlphaMissense[59]. DMS-Fold is freely and publicly available at [https://github.com/LindertLab/DMS-Fold], and includes a tutorial for usage.

## Methods

### Mega-scale targets and dataset

The mega-scale set[42] was used for discerning structural patterns from deep mutational scanning data (DMS). We used a set comprised of 175 nonredundant, structurally diverse, monomeric proteins of trimmed sequence lengths between 32 and 72 amino acids, where each protein had a structure deposited in the protein databank, exhibited near full mutational coverage, and had no sequence gaps. These proteins are characterized by high-quality protein stabilities ($\Delta\Delta$Gs) obtained from Dataset3 by Tsuboyama et al.[42]. When preparing sequences for mutational scanning, Tsuboyama et al. trimmed N- and C- terminal amino acids with a low number of contacts predicted by AlphaFold2[46]. In this work, for structural comparisons and predictions, structures and sequences were extracted from the Protein Data Bank[60] (PDB) and trimmed to correspond to their expressed sequences. Across the set, there were 9713 residues and 172295 individual mutations. Supplementary Fig. 9 shows the number of mutations per residue. There was a relative lack of mutational coverage for cysteine residues which lead us to exclude these mutations from our analysis.

### CAMEO/CASP14 set

For evaluation of DMS-Fold performance, a benchmark set was assembled which consisted of 681 proteins taken from the Continuous Automated Model EvaluatiOn (CAMEO)[54] and 29 proteins taken from the Critical Assessment of Structure Prediction (CASP14)[53] targets. PDB structures for these proteins were not trimmed. These targets have been recently deposited into the PDB and were not used in the training of AlphaFold or DMS-Fold, ensuring a rigorous assessment of DMS-Fold's predictive capacity.

### Burial metrics and burial score

Our hypothesis was that certain mutational types (e.g., small, hydrophobic to charged residue) are likely to be more destabilizing for buried residues than for surface residues. As such we needed to quantify the proximity of a residue to the surface of a protein. The per-residue surface distances of the native structures from the mega-scale set were calculated using spherical neighbor count[61], and atomic depth[62]. Spherical neighbor count assesses the number of neighboring residues within a generated sphere around a target residue. Contributions are derived from inter-residue distances between the alpha carbons of target residues and the beta carbons of neighboring residues, assigning larger contributions to smaller distances. A steepness of $1.0\,\text{\AA}^{-1}$ and a midpoint distance of 9.0 Å was used. Atomic depth calculates the depth of a residue from the surface of a protein. This metric is computed from the minimum distance between each atom of a residue and a probe traversing a simulated protein surface, where we used a surface resolution of 0.25 and a probe radius of 1.4 Å. Residue

atomic depth was averaged across the depths of each atom in the residue. Neighbor count reflects solvent exposure through residue densities while atomic depth focuses on location relative to the surface. We observed that a combination of these metrics yielded the highest possible predictive power of mutational types on protein stability. Hence, a weighted average of neighbor count and atomic depth was adopted to assess correlations between mutational $\Delta\Delta G$s and surface distance. A weight ratio of 1:2 was applied to neighbor count and atomic depth, respectively, to calculate burial extent.

The evaluation of mutational type stabilities in relation to residue surface distance across all proteins within the mega-scale set was conducted using coefficients of determination ($R^2$). This assessment involved comparing mutant $\Delta\Delta G$s to native residue wild-type burial metrics. To predict the degree of residue burial in a folded protein from folding stabilities, a metric termed "burial score" (Eq. 1) was devised. This score aims to predict the extent of residue surface distance from folding stabilities observed in the mega-scale set (Fig. 2a). Computed as a weighted average, burial score integrates the experimental $\Delta\Delta G$s of each mutation, with the weights aligned to the respective correlation coefficients ($R_i^2$). To normalize and assign greater importance to more highly correlated mutational types, a SoftMax function was applied to the set of $R_i^2$s.

$$Burial\ Score = \frac{\sum w_i \triangle\triangle G_i}{\sum w_i} = \frac{\sum Soft_{R_i^2} \triangle\triangle G_i}{\sum Soft_{R_i^2}} \tag{1}$$

### Embedding DMS in DMS-fold

A version of OpenFold[46], a PyTorch-based trainable reproduction of AlphaFold2, was modified to embed residue burial scores into the pair-representation (Fig. 2b). DMS data, through encoded burial scores, were added as an additional feature to the network. User-provided $\Delta\Delta G$s for different mutations are used to compute individual residue burial scores. These scores were encoded as integers ranging from 0 (least buried) to 9 (most buried). More negative burial scores resulted in higher integers, with burial scores less than −4.0 given a maximum score of 9. A recycling embedding layer, DMSEmbedder (Supplementary Fig. 10), was incorporated into the network, placing these encoded scores along the diagonal of a $N_{res}xN_{res}x1$ tensor. This tensor was subsequently added to the pair-representation, similar to how OpenFold recycles the MSA representation and pair-representation, during initialization followed by a linear transformation into the 128-dimensial z-space used in OpenFold. For each recycle, burial scores are embedded again into the outputted pair representation, deepening the network. MSA-subsampling[28] functionality was also integrated into DMS-Fold.

### DMS-fold training

DMS-Fold was trained and validated on a large and diverse set of proteins compiled using the PISCES[51] protein sequence culling server to ensure network generalizability, and also on 175 proteins from the mega-scale set with experimental $\Delta\Delta G$s to expose the network to actual experimental DMS-derived data. Training on only the 175 mega-scale proteins would likely have been insufficient for a network of this size. Including only entries deposited before October 2023, the Protein Data Bank (PDB) was exhaustively culled to compile nonredundant sequences with the following criteria: X-ray entries with a resolution of better than or equal to 2.5 Å, a minimum chain length of 40 residues, and a 30% identity cutoff. After excluding 9 proteins which overlapped with the mega-scale set and 144 proteins with the CASP14/CAMEO set, this curation yielded a total of 15,392 proteins. All 175 proteins from the mega-scale set were added, yielding a total of 15,567 proteins. These proteins were then randomly partitioned, allocating approximately 90% for training and 10% for validation. ThermoMPNN[52] was used to simulate changes in protein thermodynamic stabilities from

mutations. Additionally we explored an alternative model for $\Delta\Delta G$ prediction, THPLM[63] which leverages the pretrained ESM-2[4] protein language model to predict point mutation $\Delta\Delta G$s solely from a sequence. Supplementary Fig. 5 shows a comparison of the simulated $\Delta\Delta G$s from ThermoMPNN and THPLM with experimentally measured $\Delta\Delta G$s. THPLM's $\Delta\Delta G$s showed a lower correlation to experimental $\Delta\Delta G$s than those of ThermoMPNN. This was expected given that THPLM relies solely on sequence information.

Burial scores were calculated from the stabilities simulated by ThermoMPNN and were subsequently used to construct $N_{res}xN_{res}x1$ tensors for efficient embedding during the training phase. DMS-Fold, both with ThermoMPNN and THPLM derived burial scores, underwent a 40-epoch training regimen (training and validation metrics for the first 10 epochs shown in Supplementary Fig. 11, Supplementary Fig. 12) utilizing 16 NVIDIA A100 GPUs on the Ohio Supercomputer[64] Ascend cluster, initiated with DeepMind's AlphaFold2[1] 'model_5_ptm' weights as a starting point. We selected model_5_ptm since it was not trained with template information, a characteristic shared by DMS-Fold. We used OpenFold's training regimen with the default network hyper-parameters. The reduced BFD database (small_bfd) was used to generate MSAs. MSAs were uniformly subsampled with a random $N_{eff}$ between 1 and 25 to prompt the network to use the burial scores. Testing was performed with false burial scores to verify network improvements were due to burial scores and not training on shallower MSAs. DMS-Fold weights from the fifth epoch were used for inference after observing a plateau in validation metrics at 5 epochs, avoiding any potential overfitting. DMS-Fold network weights can be accessed at https://github.com/LindertLab/DMS-Fold.

### MSA-subsampling

OpenFold encodes co-evolutionary information into the network by processing Multiple Sequence Alignments (MSAs). MSA-subsampling is contingent on the specification of $N_{eff}$, the number of nonredundant sequences to be randomly subsampled from among all evolutionarily related sequences. This parameter delineates the number of sequences for inclusion in the MSA representation after being randomly shuffled. We observed that higher values of $N_{eff}$ resulted in the network relying heavily on MSAs, diminishing the influence of mutationally derived burial information on structure prediction. For the CASP14/CAMEO set (30-1411 residues), we observed a $N_{eff}$ of 10.0 to be a good balance between over- and under-reliance on the MSA. However, due to their smaller size (32-72 residues), a much smaller $N_{eff}$ (around 1.0 to 2.0) was needed for the mega-scale set. To find a balance between sequence length and DMS/MSA influence, we developed a size-dependent $N_{eff}$ which was calculated by dividing the length of the target sequence by 25.

### DMS-Fold benchmarking and analysis metrics

DMS-Fold was benchmarked on two distinct datasets: a set of CASP14 and CAMEO targets, with simulated mutational data from ThermoMPNN, and the mega-scale set with experimental DMS data. Comparative assessments were made between predictions from DMS-Fold and base OpenFold using AlphaFold2's weights. OpenFold has been shown to produce identical results to AlphaFold2[46]. For the mega-scale set, two different versions of DMS-Fold were used (see next section). To ensure deterministic predictions while benchmarking, MSA masking was disabled for both networks. Particularly in combination with a small $N_{eff}$, small variations in MSA compositions were found to lead to increased variability. For DMS-Fold, simulated and experimental stabilities, respectively were provided to the network as CSV files. Those were subsequently embedded within the pair-representation. In evaluating network performance using different $N_{eff}$ values or for examining structural characteristics of individual predictions, a single seed was used (same for both networks). However, for more comprehensive benchmarking using a

size-dependent $N_{eff}$, TM-Scores of OpenFold and DMS-Fold predictions were averaged across 25 different seeds.

The analysis metric used to measure the accuracy of predictions was template modeling score (TM-score)[65]. TM-score, used to analyze the topological similarity between a predicted and reference structure, ranges between 0.0 and 1.0, with a score of 1.0 indicating a perfect match. TM-score classifies predicted structures as having random structural similarity (0.0 < TM-score <0.17) or high fold similarity (0.5 < TM-score <1.00).

### Avoiding bias in mega-scale benchmarking with dataset splitting

The main (and user-recommended) version of DMS-Fold was trained with burial scores calculated using mutational correlations from the entire mega-scale set. To avoid any potential bias for predictions of mega-scale proteins, two alternate versions of DMS-Fold were trained using two separate subsets of the mega-scale set (Supplementary Table 2). The mega-scale set was partitioned into two subsets based on structural homology, with the aim of minimizing similarity between the subsets. Pairwise TM-Scores were calculated for all proteins in the mega-scale set and used as input for hierarchical clustering (via SciPy)[66] to separate the proteins into two structurally distinct groups (Supplementary Fig. 12). This process yielded an average intra-set TM-Score of 0.49 and an average inter-set TM-Score of 0.29. Given that TM-Scores below 0.5 typically indicate proteins are not in the same fold, this clustering approach effectively maximized structural similarity within each set while minimizing homology between them. Mutational correlation analyses were performed for each subset individually (Supplementary Fig. 2 and Fig. 3). To avoid any data leakage, ThermoMPNN was retrained using these subsets, such that proteins excluded in a subset were not included in ThermoMPNN's training and validation sets. Using these subsets, two versions of ThermoMPNN were trained. These models had similar performance during training and validation (Supplementary Fig. 14). These versions of ThermoMPNN additionally produced ΔΔGs similar to the default ThermMPNN for the mega-scale set, CASP14/CAMEO sets, and training sets (Supplementary Fig. 15). These alternate weights were then used to simulate ΔΔGs for proteins in the training set. Alternate DMS-Folds were trained using the same training set and regimen as described in previous sections (Supplementary Fig. 11), with the only difference being the exclusive use of mutational correlations derived from each subset. Predictions for each protein in the mega-scale set were generated with the version of DMS-Fold using the mega-scale subset which the protein was absent in. This ensured each mega-scale prediction was tested independently.

### Reporting summary

Further information on research design is available in the Nature Portfolio Reporting Summary linked to this article.

## Data availability

Unless otherwise stated, all data supporting the results of this study can be found in the article, supplementary, and source data files. The crystal structure data used in this study were obtained from the Protein Data Bank [https://www.rcsb.org]. Accession codes for proteins in the training, validation, and test sets are available at Huggingface [https://huggingface.co/datasets/LindertLab/]. The sequences, precomputed alignments, and CSVs with DMS data for both the CASP14/CAMEO and mega-scale sets are available at Huggingface [https://huggingface.co/datasets/LindertLab/megascale_casp14_cameo_sets]. Source data are provided with this paper. Source data for Supplementary Fig. 15 is located at Huggingface [https://huggingface.co/datasets/LindertLab/dmsfold_training_set]. Source data are provided with this paper.

## Code availability

The code for DMS-Fold and model weights are deposited at Github [https://github.com/LindertLab/DMS-Fold][67] and Zenodo [https://zenodo.org/records/15793742].

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

## Acknowledgements

We thank the members of the Lindert lab for many useful discussions and the Ohio Supercomputer Center for valuable computational resources[64]. We thank the OpenFold team for making available a fully open-source, trainable reimplementation of AlphaFold2. OpenFold can be accessed at Github [https://github.com/aqlaboratory/openfold]. Integrative protein modeling work was supported by Sloan Research Fellowship to S. L.

## Author contributions

Z.C.D. performed the code development, model training, data collection, and preparation of the manuscript along with its supplementary information. E.H.D. advised code development and provided a portion of the training set. P.D.T. and S.L. contributed to the development of the hypotheses, modeling strategy, writing of the text as well as guidance for the project.

## Competing interests

The authors declare no competing interests.
