## [Transparent Peer Review file · Nature Communications]

Deep-Learning Structure Elucidation from Single-Mutant Deep Mutational Scanning

Corresponding Author: Dr Steffen Lindert

Version 0:

Reviewer comments:

Reviewer #1

(Remarks to the Author)

The authors present DMS-Fold, a modification of OpenFold that uses predicted residual burial from a protein thermostability predictor (ThermoMPPN) or experimental deep mutational scanning (DMS) data and claim that it improves protein structure prediction.

This is potentially interesting but it seems very surprising that something as basic as predicted burial was not already learnt by alphafold, which makes us concerned about the serious possibility of data leakage and lack of appropriate splits between training and evaluation datasets.

Specifically:

[1] ThermoMPNN is based on ProteinMPNN, which is a sequence design model that uses 3D structures to generate sequences. Does DMS-fold boost the performance of alphafold for the trivial reason of data leakage: that ThermoMPNN uses structures as an input?

[2] If we understand correctly, ThermoMPNN outputs are also used to train each version of DMS-fold. ThermoMPNN was trained on mega-scale data and we can't find anything saying that the authors have removed the domains that ThermoMPNN was trained on in the evaluation.

[3] The rationale for the mega-scale dataset split isn't clear to me. Was it performed in such a way as to prevent any data leakage between proteins with homologous structures?

[4] Benchmarking DMS-fold against CASP14 & CAMEO: DMS-fold was trained on 15,536 PDB entries before October 2023. Were any CASP14 (2020) structures included in this set?

[5] Test Case with Burial Scores (7BNY). Was 7BNY part of the 15,536 training structures (it was released in 2021)?

Other queries and suggestions:

Baselines: burial and secondary structure are well predicted from sequence alone. Is the DMS data or stability prediction needed or are simpler predictors of burial or rules of thumb sufficient to improve AF predictions in the cases highlighted?

Most DMS data does not report ddGs but rather functional scores/enrichment scores, which even for abundance selections are typically related to ddGs by a non-linear (sigmoidal) function. Does this have any impact on DMS-Fold? ProteinGym and/or the Domainome DMS datasets could be used to evaluate this.

Citation of previous work: It would be helpful to the general reader if the introduction better introduced previous papers demonstrating that protein structures can be determined from DMS double mutant data (cited refs) and experimental

evolution data (<https://pubmed.ncbi.nlm.nih.gov/31838147/>). It doesn't at all distract from the current work to more precisely describe previous studies that demonstrated that DMS data can be sufficient for backbone structure determination.

"We demonstrated that residue burial information can be extracted from single-mutant deep mutational scanning (DMS)." The correlation between DMS mutational tolerance and residue depth (SASA) in the mega-scale dataset was previously reported in <https://www.biorxiv.org/content/10.1101/2023.10.06.561180v3> (and possibly other references).

It would be helpful to reduce/explain jargon and acronyms on figures. e.g. figure 1 NC and AD are not explained on the figure or in the legend.

(Remarks on code availability)

Reviewer #2

(Remarks to the Author)

I totally agree that extracting burial information from DMS data and using it for structure prediction is effective. To better understand the predictive performance of DMS-Fold, please consider the following questions:

- (1) Why does DMS-Fold use predictions from ThermoMPNN to calculate burial scores? Is it challenging to directly handle the Tsuboyama dataset?
- (2) The concept of DMS-Fold is closely related to AlphaMissense, which evaluates whether mutations are lethal. I believe DMS-Fold could potentially improve the prediction of disease-related mutations. Could you add some discussion or outlook on this aspect?
- (3) In contrast to the comment (2), has the model been evaluated on cases where mutations cause significant structural destabilization? Can it still predict the structure accurately in such cases?
- (4) Is there a reason why the DMSEmbedder matrix (Burial embedding matrix) was simply added to the pair-representation matrix?

(Remarks on code availability)

Reviewer #3

(Remarks to the Author)

Review of manuscript "Deep Learning Structure Elucidation from Single-Mutant Deep Mutational Scanning" by Drake et al.

The advent of deep learning-informed protein structure prediction is proving transformative for molecular biology. In particular, AlphaFold 2 has been widely adopted and is being used to inform experimental and evolutionary molecular biology. Nonetheless, even the most state-of-the-art approaches have limitations and can fail to produce accurate protein structure predictions for some targets, particularly those with limited sequence availability, which impacts the depth of the multiple sequence alignments used to infer structure.

One way of navigating these limitations is to integrate sources of experimental data that can be used to further guide inference. In this work, the authors explore the use of deep mutational scanning data, particularly relating to protein stability, and how this can be employed to improve the accuracy of protein structure prediction.

This work is prompted by two key concepts:

1. Certain residues, particularly hydrophobic amino acids, are likely to be buried in the three-dimensional structure of proteins, whereas polar residues are more likely to be surface exposed. From this, a metric that reflects the likely burial of an amino acid within a protein could be used to guide structure prediction.
2. Deep mutational scanning coupled to high-throughput measurements of protein stability creates a tractable system for generating the types of large datasets that could be used in such a computational framework.

The authors first use a large experimental dataset to test these foundational concepts. They demonstrate that, for example, mutation of hydrophobic residues to hydrophilic residues is likely to disrupt three-dimensional structure and therefore induce changes in thermodynamic stability. Therefore, thermodynamic stability upon mutation can be used to generate a 'burial score'.

On this basis, the authors design and implement a remix of the OpenFold deep learning architecture (OpenFold being an open-source, open training implementation of AlphaFold 2), calling this new model, DMS-Fold. This revised architecture integrates residue burial scores inferred from measurements of thermodynamic stability alongside the MSA representations as inputs for the Evoformer block of the architecture.

They train their model starting with the AlphaFold 2 weights, refining the model using simulated thermodynamic stability measurements generated with ThermoMPNN, a neural network trained to predict thermodynamic stabilities from PDB structures. The authors show that DMS-Fold can outperform AlphaFold on a validation set of proteins with simulated stability data. Through MSA subsampling, they demonstrate that the burial scores derived from deep mutational scanning data become more influential in accurate folding as the MSA becomes more shallow, suggesting that this approach may work better for difficult or divergent targets.

They then demonstrate that DMS-Fold can also outperform AlphaFold 2 for targets with matched experimental measurements of thermodynamic stability (i.e. non-simulated stability data). They also provide detailed interrogation of one particular folding target, demonstrating that the encoded burial score, derived from DMS data, correlates very well with the native burial score from the PDB structure, which in turn translates to a native-like fold in the DMS-Fold structure, but not in the AlphaFold 2 derived structure.

This is a technically well-done piece of work. The figures are excellent and clearly communicate the results. The work is novel in demonstrating that this type of experimental data can be used to improve the structure prediction, and I expect the approach to be generalisable if the target is amenable to deep mutational scanning thermodynamic measurements. The novelty, technical quality, robustness of the approach, and general interest in this area make this work attractive to a wide readership, and I am supportive of its publication.

Suggestions for improvement:

1. The authors investigate the importance of the DMS data in accurate folding through MSA subsampling. However, I wondered whether MSA masking or non-uniform changes to MSA depth could be used to examine the contribution of particular residues. For example we may hypothesise that it is particularly the buried sites that are driving the improved accuracy, therefore it may be useful to reduce the MSA-based contributions of some of these residues. I would note that this additional analysis could provide an interesting technical perspective on the method, but not fundamentally challenge the validity of the approach.

2. An important limitation is that performing deep mutational scanning and thermodynamic stability assays is not yet a trivial task. Several approaches have been developed to predict thermal stability of mutations from sequence alone (<https://pmc.ncbi.nlm.nih.gov/articles/PMC10627365/>). Can the outputs of such methods be used in combination with DMS-Fold, and how well would they perform? This would bring us back to a situation where structures can be computationally inferred from sequence alone without experimental data.

Minor suggestions:

1. There is a counterintuitive use of the phrase "burial score" where having a high burial score indicates surface exposure (based on Figure 2), while a low burial score indicates burial within the structure. This is somewhat counterintuitive. On page 11, the authors state that "residues with high predicted burial scores were generally observed to be buried in the native structure," which seems to be an opposite usage of that terminology.

2. The authors have repetitive descriptions of ThermoMPNN at the bottom of page 6 and the top of page 7. The second full explanation is not necessary.

Expertise disclosure: I have extensive experience in the application of protein structure prediction in molecular biology and to difficult target proteins. I have a good understanding of the fundamentals of how these approaches work, but I am coming from a molecular biology, not computer science perspective, and therefore am less able to provide expert comment on the architecture of the model.

(Remarks on code availability)

The code is very well documented, appears to be available in full, including the pre-trained weights via HuggingFace. I have not attempted to install run the software, however, there are instructions on how to do so.

Version 1:

Reviewer comments:

Reviewer #1

(Remarks to the Author)

I thank the authors for their responses which clarify several issues. However, I am still concerned about the way the data split was done. The authors write 'we did perform that split in a way that attempted to minimize the number of homologous structures across the two sets. This was confirmed using BLAST, yielding an average sequence identity between proteins in set 1 and set 2, of just 53.8%.' This means that there are still many proteins with homologous structures between the two sets. I think it is essential to perform the split so that there is no STRUCTURAL homology between the two sets. Otherwise there is the potential circularity that structure is being used to predict structure and the reported performance may be exaggerated.

(Remarks on code availability)

Reviewer #2

(Remarks to the Author)

My questions and concerns have been resolved. I believe the manuscript is suitable for publication.

(Remarks on code availability)

Reviewer #3

(Remarks to the Author)

The authors have done a VERY thorough job in responding to my comments and the comments of the other reviewers. I was very interested to see their extensive new analysis, and particularly the encouraging results with their integration of THPLM, which suggests sequence only analyses may be possible in the near-future. Thank you very much for a positive and highly constructive review process. I am highly supportive of publication.

I have only one minor comment. In the new text "We also observed that manually masking MSAs of residues based of off burial score had larger effects on AlphaFold2 than DMS-Fold (Supplementary Figure 7)." I think "based on burial score" would be a more appropriate and correct wording?

(Remarks on code availability)

My expertise do not allow me to provide a detailed review of the code. Nonetheless, the authors provide an up to date GitHub, with example data and a ReadMe (which possibly should be a little more verbose), and links to HuggingFace for model weights. Therefore, they provide the necessary materials for others to install and run their method.

Version 2:

Reviewer comments:

Reviewer #1

(Remarks to the Author)

I thank the authors for responding in detail to the request to minimize the sequence and structural homology between the training and test sets. Whilst I think their approach goes someway to addressing this, I don't follow why they didn't use the more standard approach of ensuring a minimum sequence / structural identity between the training and test sets. For example in the ThermoMPNN paper a 25% sequence identity cut-off is used to ensure 'none of the proteins in either Megascale or Fireprot test sets have any homologues in either training set.'

(Remarks on code availability)

Version 3:

Reviewer comments:

Reviewer #1

(Remarks to the Author)

Thank you for the clear response. The authors have now addressed my concerns.

(Remarks on code availability)

We would like to thank the reviewers for their feedback and suggested improvements. We have addressed these comments by training novel versions of ThermoMPNN used in our network training to avoid any data leakage. Additionally, we have investigated how MSAs can be modified based off burial scores, and we have explored using THPLM, a $\Delta\Delta G$ predictor which only relies on sequence information, to investigate using alternative methods of simulating data for our training set. We hope the editor and reviewers find this work fit for publication in light of these changes:

REVIEWER COMMENTS

Reviewer #1 (Remarks to the Author):

The authors present DMS-Fold, a modification of OpenFold that uses predicted residual burial from a protein thermostability predictor (ThermoMPPN) or experimental deep mutational scanning (DMS) data and claim that it improves protein structure prediction.

This is potentially interesting but it seems very surprising that something as basic as predicted burial was not already learnt by alphafold, which makes us concerned about the serious possibility of data leakage and lack of appropriate splits between training and evaluation datasets.

We agree with the reviewer that general predicted burial was definitely learned by AF2 already. However, that does not mean that DMS experiments (or any other sparse experiments) cannot yield additional protein-specific information on, for example, residue burial that go beyond (or even deviate from) the general rules that AF2 has learned from an ensemble of ~170,000 proteins. Thus, the mere fact that we see improvement upon AF2 results is by no means unexpected, nor does it immediately imply data leakage. That being said, we tried very hard to remove any possible source of data leakage, as outlined below. We originally attempted to avoid data leakage by applying our splits to the mega-scale set. By using alternate versions of DMS-Fold trained on separate subsets of mega-scale proteins, we ensure that each prediction of these proteins is completely independent of the training. As part of the revisions, we further improved our procedure by applying our splits to the training of novel versions of ThermoMPNN, removing any remaining sources of potential data leakage. The improvements observed for this dataset, we argue, are the best evidence that there is no data leakage between training and evaluation datasets.

Specifically:

[1] ThermoMPNN is based on ProteinMPNN, which is a sequence design model that uses 3D structures to generate sequences. Does DMS-fold boost the performance of alphafold for the trivial reason of data leakage: that ThermoMPNN uses structures as an input?

We appreciate the reviewer's concern regarding potential data leakage from using ThermoMPNN, which requires an input PDB to simulate DMS data. As the reviewer noted in the following comment, ThermoMPNN was trained on proteins from the mega-scale set, raising the possibility of bias in our model when predicting these same proteins.

To mitigate this, we applied mega-scale splits to ThermoMPNN and trained two separate models. These models were then used to simulate $\Delta\Delta G$ s for predicted burial scores, which were subsequently used to train our DMS-Fold models. Specifically, we generated predictions of proteins from mega-scale subset 1 using DMS-Fold and ThermoMPNN trained using subset 2, and vice versa. This dataset splitting approach ensured that the data used to simulate $\Delta\Delta G$ s was never seen during training, thereby eliminating a potential source of leakage.

We observed comparable performance between our new leakage-free ThermoMPNN models and the default ThermoMPNN during training, suggesting that any initial data leakage was minimal. This was further confirmed when we generated DMS-Fold predictions for the mega-scale sets using experimental data, which yielded very similar results. If significant data leakage had been influencing DMS-Fold's performance on CASP14/CAMEO, we would have expected a notable drop in performance when evaluating on the mega-scale set with experimental data, which would be devoid of any data leakage. However, the opposite was observed, DMS-Fold showed dramatic improvements, even surpassing the performance on CASP14/CAMEO sets. This indicates that data leakage was not a major factor in enhanced performance.

Although ThermoMPNN utilizes structural information rather than sequences as input, this does not inherently result in data leakage. The incorporation of structural data enhances ThermoMPNN's ability to predict mutational stabilities by capturing broader structural context that may not be accessible from sequence data alone. As a result, the predicted values more closely resemble realistic, experimental measurements.

Additionally, the $\Delta\Delta G$ s generated by ThermoMPNN undergo multiple processing steps before being embedded into DMS-Fold, further reducing the likelihood of leakage. First, $\Delta\Delta G$ s are averaged across all mutations for each wild-type residue, effectively combining 19 independent datapoints weighted by experimentally observed correlations. This transformation alters the original data distribution. Second, these processed burial scores are discretized into integer values, introducing further abstraction that reduces the direct influence of the original ThermoMPNN-generated values.

Based on these findings, we believe that no significant data leakage remains, as DMS-Fold's improvements persist even in the absence of ThermoMPNN-derived $\Delta\Delta G$ s, and our data preprocessing steps further reduce any potential residual effects.

“By applying dataset splitting to the training of both ThermoMPNN and DMS-Fold, we ensure that each prediction for the mega-scale set remained independent. The significant improvement

observed for the mega-scale set, combined with our use of experimentally derived burial scores, suggests no evident source of data leakage.”

[2] If we understand correctly, ThermoMPNN outputs are also used to train each version of DMS-fold. ThermoMPNN was trained on mega-scale data and we can't find anything saying that the authors have removed the domains that ThermoMPNN was trained on in the evaluation.

The reviewer makes an excellent point. In our original submission, to perform independent predictions of the mega-scale set, we randomly split the mega-scale set into two subsets. However, as the reviewer points out, ThermoMPNN was trained using the entire mega-scale set. This means there could be potential (albeit very indirect) leakage from ThermoMPNN which would make the predictions of the mega-scale set biased. To address this, we retrained ThermoMPNN after applying our mega-scale set splits. After ensuring that each training and validation set of ThermoMPNN only contained proteins from one of the subsets, rather than both, two different versions of ThermoMPNN were trained following the same regimen used by Dieckhaus et al (1). We added Supplementary Figure 12 to show the comparison in training and validation metrics between ThermoMPNN trained with the proteins from the original training sets, as well as ThermoMPNN trained with Set 1 and Set 2 proteins. This figure shows that the new ThermoMPNN models trained with proteins in either set 1 or set 2 performed similarly compared to ThermoMPNN trained with all proteins. To probe whether the new predicted $\Delta\Delta G$ s differed significantly from the $\Delta\Delta G$ s previously predicted with ThermoMPNN using the default weights, we compared predicted $\Delta\Delta G$ s from ThermoMPNN trained with proteins from Set 1 and Set 2 to the predictions using the default ThermoMPNN weights, for both the training set, mega-scale set, and the CASP14/CAMEO sets (Supplementary Figure 13). Since we saw virtually identical training behavior and high correlations between $\Delta\Delta G$ s from the default ThermoMPNN and $\Delta\Delta G$ s from the new models, we concluded that the new versions, unbiased by the proteins in our mega-scale subsets, of ThermoMPNN could predict $\Delta\Delta G$ s as accurately as the default ThermoMPNN model.

We then reperformed DMS-Fold predictions for the mega-scale set using the appropriate version of DMS-Fold trained with the corresponding, leakage-free version of simulated ThermoMPNN $\Delta\Delta G$ s. We have added these supplementary figures as well as expanded on the methods section to include the new strategy. The results for the mega-scale predictions have also been updated to reflect the use of the new DMS-Fold weights, including Figure 4. The new DMS-Fold model, trained with mega-scale splits applied to ThermoMPNN, demonstrated performance that closely matched the previous version, with only a single fewer protein showing improvement (152 vs 153) and average TM-score performance difference of just 0.01. These minimal differences suggest that using the default ThermoMPNN resulted in negligible data leakage.

“Supplementary Figure 12) Comparisons of the training and validation metrics, shown in black, blue and red respectively, of ThermoMPNN trained with all proteins, proteins from Set 1, and proteins from Set 2, respectively, across 100 epochs. a) Comparison of $\Delta\Delta G R^2$ metrics during training and validation. b) Comparison of $\Delta\Delta G$ Spearman coefficients. c) Comparison of $\Delta\Delta G$ mean squared error (MSE). d) Comparison of root mean squared error (RMSE).”

“Supplementary Figure 13) Comparison of $\Delta\Delta G$ s from ThermoMPNN using default weights and after training with Set 1 and Set 2 proteins, respectively. Color represents the density of mutation $\Delta\Delta G$ s. a) Comparison of ThermoMPNN $\Delta\Delta G$ s for the mega-scale proteins. b) Comparison of ThermoMPNN $\Delta\Delta G$ for CASP14/CAMEO proteins. c) Comparison of ThermoMPNN $\Delta\Delta G$ of proteins in DMS-Fold’s training set (excluding the mega-scale proteins).”

“To avoid any data leakage, ThermoMPNN was retrained using these subsets, such that proteins excluded in a subset were not included in ThermoMPNN’s training and validation sets. Using these subsets, two versions of ThermoMPNN were trained. The two new models had similar performance during training and validation (Supplementary Figure 12). These new versions of

ThermoMPNN additionally produced $\Delta\Delta G$ s similar to the default ThermoMPNN for the mega-scale set, CASP14/CAMEO sets, and training sets (Supplementary Figure 13). These alternate weights were then used to simulate $\Delta\Delta G$ s for proteins in the training set. Alternate DMS-Folds were trained using the same training set and regimen as described in previous sections (Supplementary Figure 10), with the only difference being the exclusive use of mutational correlations derived from each subset.”

[3] The rationale for the mega-scale dataset split isn't clear to me. Was it performed in such a way as to prevent any data leakage between proteins with homologous structures?

The rationale for the mega-scale dataset split was to ensure that we never predicted a structure of a mega-scale protein which was used in the training of the DMS-Fold network. By using a version of DMS-Fold trained on proteins from set 1 or set 2, we can generate predictions of proteins in set 2 or set 1, respectively, without any data leakage. However, while the primary purpose for the mega-scale dataset split was not to prevent data leakage between proteins with homologous structures, we did perform that split in a way that attempted to minimize the number of homologous structures across the two sets. This was confirmed using BLAST, yielding an average sequence identity between proteins in set 1 and set 2, of just 53.8%. We have added additional information in the methods describing the sequence similarity of the proteins between sets.

“The average sequence identity between proteins in set 1 and set 2 was only 53.8%, reducing the risk of data leakage between proteins with homologous structures.”

[4] Benchmarking DMS-fold against CASP14 & CAMEO: DMS-fold was trained on 15,536 PDB entries before October 2023. Were any CASP14 (2020) structures included in this set?

We thank the reviewer for their comment. We checked and realized that there were 144 proteins from the CASP14/CAMEO test set included in the original DMS-Fold training. These proteins have been removed, and DMS-Fold was retrained. This has cumulated in updated results for both the mega-scale set and CASP14/CAMEO sets (Figures 3, 4, and 5). Overall, the results for the CASP14/CAMEO sets remained largely unchanged, with only slight differences. The number of proteins that exhibited improved predictions using the DMS data decreased by five, from 636 with the original DMS-Fold to 631 with the updated version. The Average TM-score improvement remained at 0.08. Despite their initial inclusion in the training set, the removal of these proteins did not appear to significantly impact DMS-Fold's performance. We have now explicitly stated in the methods section that CASP14/CAMEO proteins were removed.

“After excluding 9 proteins which overlapped with the mega-scale set and 144 proteins which overlapped with the CASP14/CAMEO set, this curation yielded a total of 15,392 proteins. All 175 proteins from the mega-scale set were added to this training set, yielding a total of 15,567 proteins.”

[5] Test Case with Burial Scores (7BNY). Was 7BNY part of the 15,536 training structures (it was released in 2021)?

We thank the reviewer for inquiring about this. 7BNY was not among the CASP14/CAMEO proteins that were mistakenly present in the previous training set. However, as mentioned above,

we retrained DMS-Fold after removing 144 other CASP14/CAMEO proteins which overlapped with the training set. All protein structures were then re-predicted with this new version. For the 7BNY test case, the updated DMS-Fold produced an almost identical result to that of the previous version trained on the overlapping set, maintaining a TM-Score of 0.96. While the updated version of DMS-Fold yielded a meaningfully different structure for 7BNY when predicting with false burial scores (Figure 5, panel d), the overall trend remained the same, with DMS-Fold generating a completely non-native structure featuring elongated alpha-helices. So, while some of the detailed predictions changed due to the re-training, the key takeaway remains unchanged. Figure 5 has been updated to reflect results from the retrained DMS-Fold, now free of CASP14/CAMEO proteins in the training set.

Other queries and suggestions:

Baselines: burial and secondary structure are well predicted from sequence alone. Is the DMS data or stability prediction needed or are simpler predictors of burial or rules of thumb sufficient to improve AF predictions in the cases highlighted ?

We appreciate the reviewer's query, and suggest that simple predictors of residue burial are not enough to significantly improve the performance of AlphaFold2. Existing work has shown that AlphaFold2 can recognize and accurately model simple burial rules of thumb (1,2). This suggests that simple predictors are unlikely to greatly enhance the predictive capability of AlphaFold2. Other work investigating the ability of DMS-based methods to predict the effects of mutations show that DMS data can provide additional insights not available from simpler predictors, resulting in greater accuracy of predicting structural features, such residue buriedness (3,4).

(1) Bæk, K. T. & Kepp, K. P. Assessment of AlphaFold2 for Human Proteins via Residue Solvent Exposure. *Journal of Chemical Information and Modeling* 62, 3391-3400 (2022).

(2) Escobedo, N. et al. Revealing Missing Protein–Ligand Interactions Using AlphaFold Predictions. *Journal of Molecular Biology* 436, 168852 (2024).

(3) Bhasin, M. & Varadarajan, R. Prediction of Function Determining and Buried Residues Through Analysis of Saturation Mutagenesis Datasets. *Frontiers in Molecular Biosciences* 8 (2021).

(4) Chen, J., Woldring, D. R., Huang, F., Huang, X. & Wei, G. W. *Comput Biol Med* 164, 107258 (2023).

We have added the following to the manuscript to expand on the benefits of DMS data.

“Although AlphaFold2 captures underlying principles related to residue buriedness and its evolutionary significance,^{47,48} DMS data can provide a more effective and detailed understanding through empirical and context-specific insights.^{49,50”}

Most DMS data does not report ddGs but rather functional scores/enrichment scores, which even for abundance selections are typically related to ddGs by a non-linear (sigmoidal) function. Does this have any impact on DMS-Fold? ProteinGym and/or the Domainome DMS datasets could be used to evaluate this.

The reviewer presents an interesting test case to see if other related types of mutational tolerance can be used directly by DMS-Fold similar to how it currently uses $\Delta\Delta G$ s. For a subset of 13 proteins in the ProteinGym, not in the mega-scale set, and for which we have identified crystal structures, we tried directly using the reported mutational scores and calculating burial scores. We then performed predictions for 11 proteins using both DMS-Fold and AlphaFold2. However, we saw little difference in TM-Score of proteins predicted by both networks.

We suspect that most likely the nonlinear relationship between functional/enrichment scores and $\Delta\Delta G$ s are likely resulting in incorrectly or misleadingly encoded burial scores. We also suspect that poor mutational coverage from proteins in the Protein Gym set is a factor. For the mega-scale proteins, the total mutational coverage (i.e., the percentage of all possible substitution mutations for each residue) of the mega-scale proteins is 93.4%. The average mutational coverage of the 13 Protein Gym proteins was only 66.1%, a significant decrease. We included a couple of sentences in the conclusion to talk about limitations of DMS-Fold with other types of DMS data.

“Other types of DMS data (e.g. functional or enrichment scores) may potentially be compatible with DMS-Fold, assuming they correlate with $\Delta\Delta G$ s and have high mutational coverage (mega-scale proteins had 93.4% mutational coverage). Future work is required to validate this conclusion.”

Citation of previous work: It would be helpful to the general reader if the introduction better introduced previous papers demonstrating that protein structures can be determined from DMS double mutant data (cited refs) and experimental evolution data (<https://pubmed.ncbi.nlm.nih.gov/31838147/>). It doesn't at all distract from the current work to more precisely describe previous studies that demonstrated that DMS data can be sufficient for backbone structure determination.

We thank the reviewer for sharing this citation regarding other work using double mutant DMS to determine protein structure. We are referencing this paper in the manuscript now.

“Conventional, non-machine-learning protocols have made significant advances in inferring structural information of proteins from DMS and experimental evolution coupling data.³⁶⁻⁴¹”

“We demonstrated that residue burial information can be extracted from single-mutant deep mutational scanning (DMS).” The correlation between DMS mutational tolerance and residue depth (SASA) in the mega-scale dataset was previously reported in <https://www.biorxiv.org/content/10.1101/2023.10.06.561180v3> (and possibly other references).

We thank the reviewer for sharing this work and have modified the manuscript to acknowledge work done previously to compare mutational effects to structural metrics.

“Similar to previous work that showed correlations between deep mutational scanning (DMS) fitness/tolerance and solvent accessible surface area,^{57,58} we demonstrated that residue burial information can be extracted from single-mutant deep mutational scanning.”

It would be helpful to reduce/explain jargon and acronyms on figures. e.g. figure 1 NC and AD are not explained on the figure or in the legend.

The caption in Figure 1 now contains an explanation of atomic depth and neighbor count. Additionally, we have added explanations of acronyms and reduced jargon in several other figure legends.

“Figure 1) Heat maps depicting the correlation between changes in protein thermodynamic stabilities ($\Delta\Delta G$) and solubility metrics, atomic depth (AD) and neighbor count (NC), for individual mutational types across the mega-scale set. a) Comparing $\Delta\Delta G$ s to native residue atomic depth. b) Comparing $\Delta\Delta G$ s to native neighbor count. c) Differences in correlation coefficients between atomic depth and neighbor count. d) Comparing $\Delta\Delta G$ s to burial extent (defined as weighted average of both atomic depth and neighbor count).”

“Supplementary Figure 2) Heat maps depicting the correlation between changes in protein thermodynamic stabilities ($\Delta\Delta G$ s) and solubility metrics, atomic depth (AD) and neighbor count (NC), for individual mutational types across the first mega-scale subset (88 proteins). a) Comparing $\Delta\Delta G$ s to native residue atomic depth. b) Comparing $\Delta\Delta G$ s to native neighbor count. c) Differences in correlation coefficients between atomic depth and neighbor count. d) Comparing $\Delta\Delta G$ s to burial extent (defined as weighted average of both atomic depth and neighbor count).”

“Supplementary Figure 3) Heat maps depicting the correlation between changes in protein thermodynamic stabilities ($\Delta\Delta G$ s) and solubility metrics, atomic depth (AD) and neighbor count (NC), for individual mutational types across the second mega-scale subset (87 proteins). a) Comparing $\Delta\Delta G$ s to native residue atomic depth. b) Comparing $\Delta\Delta G$ s to native neighbor count. c) Differences in correlation coefficients between atomic depth and neighbor count. d) Comparing $\Delta\Delta G$ s to burial extent (defined as weighted average of both atomic depth and neighbor count).”

“Figure 3) Performance of DMS-Fold on the CASP14/CAMEO targets set with simulated changes in protein thermodynamic stabilities ($\Delta\Delta G$ s). a) Template modeling score (TM-Score) comparison of predictions from DMS-Fold and AlphaFold2 using a size-dependent number of nonredundant sequences (N_{eff}). Size of each marker represents the N_{eff} used for MSA subsampling. Color represents the change in network confidence, pLDDT, between DMS-Fold and AlphaFold2. b) TM-Score distributions of both networks binned to TM-Scores of AlphaFold2 predictions. c) TM-Score distributions of predictions from both DMS-Fold and AlphaFold2 using different uniform N_{eff} values. d) Five predicted structures (aligned to native structure (grey)) where DMS-Fold with a size-dependent N_{eff} (blue) had a TM-Score improvement > 0.5 compared to AlphaFold2 with no MSA-subsampling (orange). e) Comparison of changes in pLDDTs and TM-Scores between predictions with DMS-Fold and AlphaFold2. Color represents the change in the difference of solubility metrics for the DMS-Fold structure and the native structure with the AlphaFold2 structure and the native structure.”

“Figure 4) Performance of DMS-Fold on the Mega-scale set using experimental changes in protein thermodynamic stabilities ($\Delta\Delta G$ s). a) Template modeling score (TM-Score) comparison of predictions from DMS-Fold and AlphaFold2 using a size-dependent number of nonredundant sequences (N_{eff}). Size of each marker represents the N_{eff} used for MSA subsampling. Color represents the change in network confidence, pLDDT between DMS-Fold and AlphaFold2. b) TM-Score distributions of both networks binned to TM-Scores of AlphaFold2 predictions. c) Top five predicted structures from AlphaFold2 with a size-dependent N_{eff} (orange) and DMS-Fold with a size-dependent N_{eff} (blue) aligned to their native structure (grey). d) TM-Score distributions of predictions from both DMS-Fold and AlphaFold2 using different uniform N_{eff} values. e) Comparison of changes in pLDDTs and TM-Scores between predictions with DMS-Fold and AlphaFold2. Color represents the change in the difference of solubility metrics for the DMS-Fold structure and the native structure with the AlphaFold2 structure and the native structure.”

“Supplementary Figure 10) Training and validation metrics of DMS-Fold for the main model, mega-scale split with subset 1, and subset 2. a) Subset of DMS-Fold training metrics, including training loss per-epoch, distance root mean squared deviation with respect to the alpha carbon (dRMSD_CA) per-epoch, local distance difference test with respect to the alpha carbon (IDDT_CA) per-epoch, and frame aligned point error (FAPE) loss per-epoch. b) Subset of DMS-Fold validation metrics, including validation loss per-epoch, dRMSD_CA per-epoch, IDDT_CA per-epoch, and FAPE loss per-epoch.”

Reviewer #2 (Remarks to the Author):

I totally agree that extracting burial information from DMS data and using it for structure prediction is effective. To better understand the predictive performance of DMS-Fold, please consider the following questions:

We thank the reviewer for their comment and support of the premise behind our work.

(1) Why does DMS-Fold use predictions from ThermoMPNN to calculate burial scores? Is it challenging to directly handle the Tsuboyama dataset?

We appreciate the reviewer's question. Since AlphaFold2 was trained using almost the entirety of the Protein Data Bank, we wanted to train our network on a sufficiently large set of diverse proteins to ensure that DMS-Fold can predict a wide range of proteins. We culled the PDB for 15,392 proteins, but since we did not have DMS data for these proteins, we simulated mutation $\Delta\Delta G$ s using ThermoMPNN. When training DMS-Fold, we then included the Tsuboyama proteins (with burial scores calculated from the experimental $\Delta\Delta G$ s) for a training set of 15,567 proteins. Notably, training on only the small set of 175 proteins of known structure in the Tsuboyama dataset would likely have led to DMS-Fold having difficulty learning the relationship between burial scores and residue core proximity and would result in the network being less generalizable. We've expanded on our explanation of DMS-Fold's training set in the method section of the manuscript:

“DMS-Fold was trained and validated on a large and diverse set of proteins compiled using the PISCES⁵¹ protein sequence culling server to ensure network generalizability, and also on 175 proteins from the mega-scale set with experimental $\Delta\Delta G$ s to expose the network to actual experimental DMS-derived data. Training on only the 175 mega-scale proteins would likely have been insufficient for a network of this size.”

(2) The concept of DMS-Fold is closely related to AlphaMissense, which evaluates whether mutations are lethal. I believe DMS-Fold could potentially improve the prediction of disease-related mutations. Could you add some discussion or outlook on this aspect?

The reviewer brings up an interesting point. DMS-Fold was developed for the purpose of using structural information from single mutant DMS to help guide AlphaFold2 structural inference. Because of this, the DMS data is processed into scores only used to predict structural properties of the wild-type sequence to more accurately predict its structure, rather than predict the structure of mutated sequences. DMS-Fold could be expanded to include mutational types and predict the structural effects of deleterious mutations, and this would be an interesting avenue for future work. The following section has been added within the discussion section of the manuscript to discuss possible future work in using DMS-Fold to identify lethal mutations.

“Although DMS-Fold currently only supports the use of mutagenic data for predicting structures of wild type sequences, future work could focus on expanding the network to predict the structural effects of deleterious mutations or even identify lethal mutations, similar to AlphaMissense.⁵⁹”

(3) In contrast to the comment (2), has the model been evaluated on cases where mutations cause significant structural destabilization? Can it still predict the structure accurately in such cases?

We thank the reviewer for their comment. The Tsuboyama dataset contains thermodynamic folding stabilities of nearly all possible point mutations of hundreds of proteins, with a significant portion being structurally destabilizing. A total of 172,295 mutations are available across the 175 proteins

from the Tsuboyama dataset used to train and evaluate DMS-Fold. 50.9% of these mutations had $\Delta\Delta Gs > -0.5$ kcal/mol, 40.9% had $-0.5 > \Delta\Delta Gs > -2.5$ kcal/mol, and 8.2% had $\Delta\Delta Gs < -2.5$ kcal/mol. As a result, DMS-Fold inference on these 175 proteins from the Tsuboyama dataset would result in DMS-Fold being evaluated on a significant number of destabilizing mutations, which would be encoded into burial scores. To investigate the impact of high encoded burial scores (buried residues with destabilizing mutations) on the accuracy of predicted structures, we looked at how residue root-mean-squared deviations (RMSDs) to the native structure of each protein compared to the residue burial score, calculated from experimental $\Delta\Delta Gs$. Supplementary Figure 6 compares averaged per-residue RMSDs across all 175 predicted DMS-Fold structures, which are also averaged across all 25 seeds, to experimentally derived burial scores (both encoded and unencoded).

From this analysis, we can see that most residues (independent of burial scores) were modeled accurately with DMS-Fold, but there was less variation in accuracy for residues predicted to be more buried (large negative unencoded BS, large positive encoded BS). The maximum averaged per-residue RMSDs observed for residues with encoded burial scores of 0 and 1 were 39.2 Å and 44.5 Å, respectively, while the maximum RMSDs for residues with larger encoded burial scores of 8 and 9 were 3.0 Å and 3.3 Å, respectively. This suggests that buried residues with highly destabilizing mutations, i.e. larger encoded burial scores, were likely to be modeled more accurately. We have added the following sentences to the manuscript discussing the effect of destabilizing mutations on DMS-Fold and this subsequent analysis in the supporting information.

“Supplementary Figure 6) Comparison of averaged per-residue α -carbon root-mean-squared deviations (CA RMSDs) of DMS-Fold predicted structures and experimentally derived burial scores of mega-scale proteins. RMSDs were averaged across predictions from 25 seeds a) Scatter plot of residue RMSDs vs unencoded burial scores. b) Boxplots of residue RMSDs for different encoded burial scores.”

“It was observed that residues with larger encoded burial scores generally were more likely to be modeled more accurately. This can be seen in Supplementary Figures 6a and 6b, where residues with low encoded burial scores of 0 and 1 had greater variability of α -carbon (CA) per-residue root-mean-squared deviations (RMSD) with maximum values of 39.2 Å and 44.5 Å, respectively

while higher encoded burial scores of 8 and 9 resulted in maximum RMSDs of 3.0 Å and 3.3 Å. This underscores that the information encoded in the burial score (i.e. the fact that a residue is located in a buried location in the core of the protein) allowed us to more accurately restrain those residues to their correct positions.”

(4) Is there a reason why the DMSEmbedder matrix (Burial embedding matrix) was simply added to the pair-representation matrix?

We thank the reviewer for their comment. The DMS embeddings are added to the pair-representation simply to adhere to similar operations within AlphaFold2, such as the addition of the recycled MSA and pair-representations from a previous cycle. Also, the addition of the embeddings is the simplest way to integrate DMS restraints. For example, the matrix could have been multiplied to the pair representation, but this could have resulted in overemphasis of certain experimental constraints from the DMS. We add some additional explanation regarding how the embeddings are combined with the pair-representation. Additionally, we added an SI Figure clearly describing the detailed matrix addition.

“A new recycling embedding layer, DMSEmbedder (Supplementary Figure 9), was incorporated into the network, placing these encoded scores along the diagonal of a $N_{res} \times N_{res} \times 1$ tensor. This tensor was subsequently added to the pair-representation, similar to how OpenFold recycles the MSA representation and pair-representation, during initialization followed by a linear transformation into the 128-dimensional z-space used in OpenFold.”

“Supplementary Figure 9) Diagram depicting the embedding of the burial scores into DMS-Fold’s pair representation. An initial $N_{res} \times N_{res}$ is constructed where burial scores are placed along the diagonal. A new linear layer transforms this tensor to match the dimensionality of OpenFold’s pair representation, resulting in a $N_{res} \times N_{res} \times 128$ tensor. The embedded tensor is then added to the pair representation.”

Reviewer #3 (Remarks to the Author):

Review of manuscript "Deep Learning Structure Elucidation from Single-Mutant Deep Mutational Scanning" by Drake et al.

The advent of deep learning-informed protein structure prediction is proving transformative for molecular biology. In particular, AlphaFold 2 has been widely adopted and is being used to inform experimental and evolutionary molecular biology. Nonetheless, even the most state-of-the-art approaches have limitations and can fail to produce accurate protein structure predictions for some targets, particularly those with limited sequence availability, which impacts the depth of the multiple sequence alignments used to infer structure.

One way of navigating these limitations is to integrate sources of experimental data that can be used to further guide inference. In this work, the authors explore the use of deep mutational scanning data, particularly relating to protein stability, and how this can be employed to improve the accuracy of protein structure prediction.

This work is prompted by two key concepts:

1. Certain residues, particularly hydrophobic amino acids, are likely to be buried in the three-dimensional structure of proteins, whereas polar residues are more likely to be surface exposed. From this, a metric that reflects the likely burial of an amino acid within a protein could be used to guide structure prediction.

2. Deep mutational scanning coupled to high-throughput measurements of protein stability creates a tractable system for generating the types of large datasets that could be used in such a computational framework.

The authors first use a large experimental dataset to test these foundational concepts. They demonstrate that, for example, mutation of hydrophobic residues to hydrophilic residues is likely to disrupt three-dimensional structure and therefore induce changes in thermodynamic stability. Therefore, thermodynamic stability upon mutation can be used to generate a 'burial score'.

On this basis, the authors design and implement a remix of the OpenFold deep learning architecture (OpenFold being an open-source, open training implementation of AlphaFold 2), calling this new model, DMS-Fold. This revised architecture integrates residue burial scores inferred from measurements of thermodynamic stability alongside the MSA representations as inputs for the Evoformer block of the architecture.

They train their model starting with the AlphaFold 2 weights, refining the model using simulated thermodynamic stability measurements generated with ThermoMPNN, a neural network trained to predict thermodynamic stabilities from PDB structures. The authors show that DMS-Fold can outperform AlphaFold on a validation set of proteins with simulated stability data. Through MSA subsampling, they demonstrate that the burial scores derived from deep mutational scanning data become more influential in accurate folding as the MSA becomes more shallow, suggesting that this approach may work better for difficult or divergent targets.

They then demonstrate that DMS-Fold can also outperform AlphaFold 2 for targets with matched experimental measurements of thermodynamic stability (i.e. non-simulated stability data). They also provide detailed interrogation of one particular folding target, demonstrating that the encoded burial score, derived from DMS data, correlates very well with the native burial score from the PDB structure, which in turn translates to a native-like fold in the DMS-Fold structure, but not in the AlphaFold 2 derived structure.

This is a technically well-done piece of work. The figures are excellent and clearly communicate the results. The work is novel in demonstrating that this type of experimental data can be used to improve the structure prediction, and I expect the approach to be generalisable if the target is amenable to deep mutational scanning thermodynamic measurements. The novelty, technical quality, robustness of the approach, and general interest in this area make this work attractive to a wide readership, and I am supportive of its publication.

We thank the reviewer for their detailed summary and the positive assessment.

Suggestions for improvement:

1. The authors investigate the importance of the DMS data in accurate folding through MSA subsampling. However, I wondered whether MSA masking or non-uniform changes to MSA depth could be used to examine the contribution of particular residues. For example we may hypothesise that it is particularly the buried sites that are driving the improved accuracy, therefore it may be useful to reduce the MSA-based contributions of some of these residues. I would note that this additional analysis could provide an interesting technical perspective on the method, but not fundamentally challenge the validity of the approach.

The reviewer makes an intriguing point, and we agree that residue-based MSA masking could be a promising way to investigate the effects of encoded burial scores on residues at buried sites. We took the mega-scale set and did predictions using both DMS-Fold and OpenFold where the MSAs were modified based off encoded burial scores. For residues that were predicted to be buried, defined using a burial score cutoff, MSAs were modified by replacing the residue identity in the sequence with 'X', indicating the identity of that residue is unknown. Structures were generated using these modified MSAs for proteins in the mega-scale set using DMS-Fold and OpenFold.

Supplementary Figure 7 was added to show the range in TM-scores of both networks with these modified MSAs at different cutoffs. Using a lower encoded burial score cutoff, meaning replacing all highly buried residues and most slightly buried residues with 'X', resulted in dramatically worse predictions from OpenFold. Interestingly, DMS-Fold recovered quite significantly, despite the highly masked MSAs. This suggests that the information from the DMS is enough to compensate from the lack of defined residues in aligned sequences for buried residues. As this cutoff was raised, meaning only very buried residues were masked, OpenFold's and DMS-Fold's predictions improved overall, but with DMS-Fold being much more consistent. We have added Supplementary Figure 7. This interesting analysis showed that masking residues in the MSAs based off burial scores is another valid subsampling technique. Additionally, it showed that DMS-Fold was still able to recover for a large portion of proteins with only the inclusion of burial scores, adding an interesting technical perspective on the method.

“Supplementary Figure 7) Boxplot comparing the TM-Score distributions of DMS-Fold and AlphaFold2 with custom-modified MSAs based off an encoded burial score cutoff. After MSAs were generated for a particular target sequence, these MSAs were modified such that buried residues, meaning residues with an encoded burial score greater than the specified cutoff, were replaced with 'X'. Blue indicates DMS-Fold predictions and orange indicates AlphaFold2 predictions.”

“We also observed that manually masking MSAs of residues based off burial score had larger effects on AlphaFold2 than DMS-Fold (Supplementary Figure 7).”

2. An important limitation is that performing deep mutational scanning and thermodynamic stability assays is not yet a trivial task. Several approaches have been developed to predict

thermal stability of mutations from sequence alone (<https://pmc.ncbi.nlm.nih.gov/articles/PMC10627365/>). Can the outputs of such methods be used in combination with DMS-Fold, and how well would they perform? This would bring us back to a situation where structures can be computationally inferred from sequence alone without experimental data.

Accurately predicting the thermal stabilities of mutations from sequence alone would be highly beneficial. ThermoMPNN requires structural input for inference, which may limit its applicability. As suggested by the reviewer, we explored using THPLM, which leverages transfer learning with ESM2 to estimate point mutation thermal stabilities. We calculated burial scores from THPLM $\Delta\Delta G$ s and generated new DMS-Fold predictions for the mega-scale proteins. The following figure depicts the TM-score comparisons between AlphaFold2 and DMS-Fold using ThermoMPNN and THPLM derived burial scores, respectively. When comparing AlphaFold2 with ThermoMPNN-based DMS-Fold we observed that 152 proteins showed improved DMS-Fold predictions with an average TM-Score increase of 0.17 (left panel). When comparing AlphaFold2 with THPLM-based DMS-Fold, we observed mixed results: 78 proteins showed improved DMS-Fold predictions using THPLM burial scores, with an average TM-score increase of 0.13, while 97 proteins had worse predictions than AlphaFold2, with an average TM-score decrease of 0.13 (right panel).

Further investigation suggests that the poorer performance (as compared to using ThermoMPNN derived burial scores, Figure 4, and left panel above) is likely due to the lower accuracy of predicted stabilities using THPLM. Supplementary Figure 5 was added which compares ThermoMPNN, THPLM, and experimental stabilities for the mega-scale set. As expected, ThermoMPNN predictions are significantly more accurate than those of THPLM (when compared to the experimental stabilities). Additionally, there is greater variance between ThermoMPNN and THPLM stabilities. The lower quality of THPLM stabilities likely contributes to poorer performance for some proteins by providing less accurate structural information, ultimately affecting the reliability of DMS-Fold predictions.

“Supplementary Figure 5) Comparison of simulated $\Delta\Delta G_s$ from ThermoMPNN and THPLM with experimentally measured $\Delta\Delta G_s$ for all available point mutations in the mega-scale set. a) ThermoMPNN-simulated $\Delta\Delta G_s$ versus experimental $\Delta\Delta G_s$. b) THPLM-simulated $\Delta\Delta G_s$ versus experimental $\Delta\Delta G_s$. c) Comparison of $\Delta\Delta G_s$ simulated by ThermoMPNN and THPLM.”

“Supplementary Figure 5 shows a comparison of the simulated $\Delta\Delta G_s$ from ThermoMPNN and THPLM with experimentally measured $\Delta\Delta G_s$. THPLM’s $\Delta\Delta G_s$ showed a lower correlation to experimental $\Delta\Delta G_s$ than those of ThermoMPNN. This was expected given that THPLM relies solely on sequence information.”

However, it could be argued that inputting THPLM-predicted stabilities into a network trained on ThermoMPNN-predicted stabilities may not provide a fair basis for comparison. We hypothesized that training DMS-Fold using THPLM stabilities could help the model better accommodate less accurate stability predictions. To test this, we generated stability predictions for each point mutation across the 15,567 proteins in DMS-Fold’s training set, yielding approximately 73 million mutations. Burial scores were calculated, and DMS-Fold was trained following the same protocol as the original model. SI Figure 11 shows the training and validation metrics of these new models. We then did predictions for proteins in the mega-scale set. As in our previous approach for the mega-scale set, we applied data splits to train two distinct DMS-Fold models for these predictions.

Since THPLM was originally trained on datasets from ProTherm, retraining it with the mega-scale splits was unnecessary.

“Supplementary Figure 11) Training and validation metrics of DMS-Fold trained using THPLM-derived burial scores for the main model, mega-scale split with subset 1, and subset 2. a) Subset of DMS-Fold training metrics, including training loss per-epoch, distance root mean squared deviation with respect to the alpha carbon (dRMSD_CA) per-epoch, local distance difference test with respect to the alpha carbon (IDDT_CA) per-epoch, and frame aligned point error (FAPE) loss per-epoch. b) Subset of DMS-Fold validation metrics, including validation loss per-epoch, dRMSD_CA per-epoch, IDDT_CA per-epoch, and FAPE loss per-epoch.”

“Additionally we explored an alternative model for $\Delta\Delta G$ prediction, THPLM⁶³ which leverages the pretrained ESM-2⁴ protein language model to predict point mutation $\Delta\Delta G$ s solely from a sequence.”

“DMS-Fold, both with ThermoMPNN and THPLM derived burial scores, underwent a 40-epoch training regimen (training and validation metrics for the first 10 epochs shown in Supplementary Figure 10, Supplementary Figure 11) utilizing 16 NVIDIA A100 GPUs on the Ohio Supercomputer⁶⁴ Ascend cluster, initiated with DeepMind’s AlphaFold2¹ ‘model_5_ptm’ weights as a starting point.”

For the mega-scale targets, DMS-Fold trained with THPLM stabilities yielded more accurate predictions for 121 proteins, with an average TM-score improvement of 0.09. In comparison, DMS-Fold trained with ThermoMPNN produced better structures for 152 proteins, achieving a higher average improvement of 0.17. While both networks demonstrate significant enhancements in protein structure with DMS-Fold, the ThermoMPNN-trained model performed notably better. Since each mega-scale prediction was independent of DMS-Fold’s training, and experimental $\Delta\Delta G$ s were used to predict burial scores, the weaker performance of the THPLM-based model can likely be attributed to the lower quality of its predicted stabilities.

Overall, THPLM is an impressive model for predicting point mutation stabilities, and future iterations will likely further refine its predictive capabilities. Although DMS-Fold trained with THPLM stabilities underperformed compared to the ThermoMPNN-based model, a significant number of targets saw an improvement, suggesting that DMS-Fold has potential uses with an exclusively sequence-based predictor of point mutation stabilities.

“**Supplementary Figure 4)** Performance of DMS-Fold trained using THPLM protein thermodynamic stabilities ($\Delta\Delta G$ s) on the mega-scale targets with experimental $\Delta\Delta G$ s. a) Template modeling score (TM-Score) comparison of predictions from DMS-Fold and AlphaFold2 using a size-dependent number of nonredundant sequences (N_{eff}). Size of each marker represents the N_{eff} used for MSA subsampling. Color represents the change in network confidence, pLDDT, between DMS-Fold and AlphaFold2.”

“Supplementary Figure 4 shows the comparison of AlphaFold2 to DMS-Fold trained with burial scores derived from THPLM, which resulted in 121 proteins improving with an average TM-score increase of only 0.09. The poorer performance of this DMS-Fold model is most likely due to the difficulty of accurately predicting stabilities from only sequence information (as demonstrated in Supplementary Figure 5).”

“We observed improvements with an alternative model, THPLM, which relies solely on protein sequences to simulate point mutation stabilities. Although DMS-Fold trained with THPLM stabilities performed worse than the ThermoMPNN-based model, a notable number of the mega-scale targets showed improvement, indicating that DMS-Fold may be useful when paired with a sequence-based predictor of point mutation stabilities.”

Minor suggestions:

1. There is a counterintuitive use of the phrase "burial score" where having a high burial score indicates surface exposure (based on Figure 2), while a low burial score indicates burial within the structure. This is somewhat counterintuitive. On page 11, the authors state that "residues with high predicted burial scores were generally observed to be buried in the native structure," which seems to be an opposite usage of that terminology.

We apologize for any confusion regarding our terminology for burial scores and thank the reviewer for bringing up the issue. We have made some modifications to the manuscript where we explicitly differentiate between "encoded burial scores", where higher values indicate residues predicted to be buried, and "unencoded burial scores" which represent weighted and averaged $\Delta\Delta G$ s, meaning that lower values indicate residues predicted to be buried. We also changed Figure 2 to be clearer and now use the correct label.

"We used two types of burial scores in this study: "encoded burial scores", where higher values indicate residues predicted to be buried, and "unencoded burial scores" which represent weighted and averaged $\Delta\Delta G$ s, meaning that lower values indicate residues predicted to be buried."

"DMS data was added as an additional feature to the network as well as a new recycle embedder which embedded these encoded burial scores along the diagonal of the pair representation during initialization prior to Evoformer processing (Figure 2b)."

"In Figure 5c, per-residue encoded burial scores and burial extents are compared between the native crystal structure, the DMS-Fold prediction, and the AlphaFold2 prediction. Residues with high predicted encoded burial scores were generally observed to be buried in the native structure, as seen from agreement with burial extents."

"DMS data, through encoded burial scores, were added as an additional feature to the network."

a

Predicting Burial Scores Using Structurally Disruptive Mutation Types

**b**
2. The authors have repetitive descriptions of ThermoMPNN at the bottom of page 6 and the top of page 7. The second full explanation is not necessary.

We agree with the reviewer that the second explanation of ThermoMPNN being a graph neural network trained on the mega-scale dataset was redundant. We have removed the explanation and modified the sentences to reflect this. We also removed another explanation in the methods.

“Folding stabilities of point mutations were simulated for these proteins using ThermoMPNN⁵².”

“ThermoMPNN⁵² was used to simulate changes in protein thermodynamic stabilities from mutations.”

Expertise disclosure: I have extensive experience in the application of protein structure prediction in molecular biology and to difficult target proteins. I have a good understanding of the fundamentals of how these approaches work, but I am coming from a molecular biology, not computer science perspective, and therefore am less able to provide expert comment on the architecture of the model.

Reviewer #3 (Remarks on code availability):

The code is very well documented, appears to be available in full, including the pre-trained weights via HuggingFace. I have not attempted to install run the software, however, there are instructions on how to do so.

We would like to thank the reviewers for their feedback and suggested improvements. In response, we have applied hierarchical clustering to divide the mega-scale dataset into two subsets with no structural homology between them. This clustering was performed using pairwise TM-Scores calculated with TM-Align. Subsequently, we retrained the ThermoMPNN and DMS-Fold models and regenerated the mega-scale predictions, and observed results consistent with what was previously reported. We hope that these revisions address the reviewer's concerns, and that the manuscript is now suitable for publication.

REVIEWER COMMENTS

Reviewer #1 (Remarks to the Author):

I thank the authors for their responses which clarify several issues. However, I am still concerned about the way the data split was done. The authors write 'we did perform that split in a way that attempted to minimize the number of homologous structures across the two sets. This was confirmed using BLAST, yielding an average sequence identity between proteins in set 1 and set 2, of just 53.8%.' This means that there are still many proteins with homologous structures between the two sets. I think it is essential to perform the split so that there is no STRUCTURAL homology between the two sets. Otherwise there is the potential circularity that structure is being used to predict structure and the reported performance may be exaggerated.

We thank the reviewer for their thoughtful concern, and we agree that having high homology between the sets would be problematic. Upon further examination of our BLAST analysis, we realized that the reported identity values were inaccurate (too high). Specifically, BLAST does not generate alignments for sequences that are too dissimilar, and as a result, these were excluded from our analysis. Consequently, our reported average only included sequences for which BLAST returned alignments, which was not our original intention. We apologize for this error. While these corrected sequence identity numbers would have likely allayed the reviewer's concerns about data leakage, the previous split did not represent the minimum possible structural homology between the two sets.

Thus, in the spirit of fully addressing the reviewer's concern regarding minimizing structural homology between the sets, we performed a comprehensive analysis using TM-align. We computed TM-Scores for all pairwise comparisons in the mega-scale dataset and then clustered the proteins based on these scores. Below, we include a pairwise heatmap of TM-Scores for all proteins in the mega-scale set, organized according to the original dataset splits. Proteins from set 1 (88 proteins) are ordered first along both axes, followed by those in set 2 (87 proteins). This visualization provides a clearer view of the structural similarity between the two sets.

From the distribution of TM-Scores in the above heatmap, we observed a substantial number of structurally homologous protein pairs between the original sets (TM-Scores > 0.5 in the off-diagonal squares). The number of structurally homologous proteins between sets was 822, and the number within both sets was 886. To reduce structural homology between the sets, we applied hierarchical clustering (using SciPy) based on pairwise TM-Scores. This clustering produced two distinct groups: one containing 35 proteins and the other 140 proteins. SI Figure 12 shows a pairwise heatmap of TM-Scores for all proteins, now organized according to these newly defined, structurally distinct sets.

“Supplementary Figure 12) Pairwise heatmap of TM-Scores between proteins in the mega-scale set. Proteins are grouped and ordered along the axes according to their set designation (Set 1: 35

proteins; Set 2: 140 proteins), with boundaries indicated by black lines. The color scale represents the TM-Score, with warmer colors indicating higher structural similarity.”

Although the newly defined sets are no longer of similar size, they exhibit substantially reduced structural homology compared to the original split. Previously, the average intra-set TM-Score was 0.33, and the average inter-set TM-Score was 0.329. In contrast, the new clustering-based split yields an average intra-set TM-Score of 0.49 and average inter-set TM-Score of 0.29, reflecting a significant reduction in structural similarity between the two sets. Using these newly defined sets, we repeated all downstream analyses in the DMS-Fold pipeline for the mega-scale predictions, including deriving new mutational correlations, ThermoMPNN model trainings, training set generation, and DMS-Fold training. This produced results that were highly consistent with our previous findings, as shown in the updated Figure 4.

“Figure 4) Performance of DMS-Fold on the Mega-scale set using experimental changes in protein thermodynamic stabilities ($\Delta\Delta G$ s). a) Template modeling score (TM-Score) comparison of predictions from DMS-Fold and AlphaFold2 using a size-dependent number of nonredundant sequences (N_{eff}). Size of each marker represents the N_{eff} used for MSA subsampling. Color represents the change in network confidence, pLDDT between DMS-Fold and AlphaFold2. b) TM-Score distributions of both networks binned to TM-Scores of AlphaFold2 predictions. c) Top five predicted structures from AlphaFold2 with a size-dependent N_{eff} (orange) and DMS-Fold with a size-dependent N_{eff} (blue) aligned to their native structure (grey). d) TM-Score distributions of predictions from both DMS-Fold and AlphaFold2 using different uniform N_{eff} values. e) Comparison of changes in pLDDTs and TM-Scores between predictions with DMS-Fold and AlphaFold2. Color represents the change in the difference of solubility metrics for the DMS-Fold structure and the native structure with the AlphaFold2 structure and the native structure.”

The number of mega-scale targets that showed improvement using the newly trained DMS-Fold models was 149, closely matching the previously reported number of 152 (see panel a). The average improvement in TM-Score remained consistent at 0.17, while the number of targets with TM-Score improvement > 0.1 increased from 86 to 90. To further assess potential data leakage from structural homology, we also evaluated the impact of excluding mega-scale proteins entirely from the DMS-Fold training set. Using the new dataset splits for correlation analyses, we retrained DMS-Fold without any mega-scale proteins in the training data. SI Figure 8 shows the results of this evaluation. The findings were consistent with previous results, with 149 targets showing improvement and an average TM-Score increase of 0.18. These results provide additional strong evidence that there was no data leakage from including the mega-scale proteins in training DMS-Fold.

“DMS-Fold predictions were generated through the application of dataset splitting (see methods) to minimize structural homology between the sets and to ensure each prediction avoids any bias or data leakage.”

“The mega-scale set was partitioned into two subsets based on structural homology, with the aim of minimizing similarity between the subsets. Pairwise TM-Scores were calculated for all proteins in the mega-scale set and used as input for hierarchical clustering (via SciPy) to separate the proteins into two structurally distinct groups (Supplementary Figure 12). This process yielded an average intra-set TM-Score of 0.49 and an average inter-set TM-Score of 0.29. Given that TM-Scores below 0.5 typically indicate proteins are not in the same fold, this clustering approach effectively maximized structural similarity within each set while minimizing homology between them.”

In summary, we hope that our revised mega-scale dataset splitting, using hierarchical clustering based on TM-Scores to minimize structural homology between sets, along with the consistency of our updated results, provides convincing evidence to the reviewer that our findings are not impacted by data leakage. We further believe that this reinforces the potential of DMS-guided deep learning as a powerful approach for protein structure prediction in structural biology.

“Supplementary **Figure 8**) Performance of DMS-Fold trained without the Mega-scale proteins, on the Mega-scale set using experimental changes in protein thermodynamic stabilities ($\Delta\Delta G$ s). a) Template modeling score (TM-Score) comparison of predictions from DMS-Fold and AlphaFold2 using a size-dependent number of nonredundant sequences (N_{eff}). Size of each marker represents the N_{eff} used for MSA subsampling. Color represents the change in network confidence, $pLDDT$ between DMS-Fold and AlphaFold2. b) TM-Score distributions of both networks binned to TM-Scores of AlphaFold2 predictions. c) TM-Score distributions of predictions from both DMS-Fold and AlphaFold2 using different uniform N_{eff} values. d) Comparison of changes in $pLDDT$ s and TM-Scores between predictions with DMS-Fold and AlphaFold2. Color represents the change in the difference of solubility metrics for the DMS-Fold structure and the native structure with the AlphaFold2 structure and the native structure.”

“Although the mega-scale set was split into two structurally distinct groups, we investigated potential data leakage from including the mega-scale proteins at all in the DMS-Fold training set. To address this, we retrained DMS-Fold excluding all mega-scale proteins from the training data. As shown in Supplementary Figure 8, predictions on the mega-scale set using the new models were consistent with the results presented in Figure 4, leading us to conclude that no data leakage from including the mega-scale proteins was present.”

Reviewer #2 (Remarks to the Author):

My questions and concerns have been resolved. I believe the manuscript is suitable for publication.

We sincerely thank the reviewer for their thoughtful feedback and for recognizing the improvement to the manuscript. We appreciate their time and effort and are pleased to hear that they find the manuscript suitable for publication.

Reviewer #3 (Remarks to the Author):

The authors have done a VERY thorough job in responding to my comments and the comments of the other reviewers. I was very interested to see their extensive new analysis, and particularly the encouraging results with their integration of THPLM, which suggests sequence only analyses may be possible in the near-future. Thank you very much for a positive and highly constructive review process. I am highly supportive of publication.

I have only one minor comment. In the new text “We also observed that manually masking MSAs of residues based of off burial score had larger effects on AlphaFold2 than DMS-Fold (Supplementary Figure 7).” I think "based on burial score" would be a more appropriate and correct wording?

We thank the reviewer for their generous and encouraging feedback. We greatly appreciate their excitement and are pleased the new analyses and findings were of interest. Regarding the minor comment, we agree with the suggested wording change and have updated the text.

“We also observed that manually masking MSAs of residues based on burial score had larger effects on AlphaFold2 then DMS-Fold (Supplementary Figure 7).”

Reviewer #3 (Remarks on code availability):

My expertise do not allow me to provide a detailed review of the code. Nonetheless, the authors provide an up to date GitHub, with example data and a ReadMe (which possibly should be a little more verbose), and links to HuggingFace for model weights. Therefore, they provide the necessary materials for others to install and run their method.

We updated the GitHub with more verbose instructions and descriptions.

REVIEWER COMMENTS

Reviewer #1 (Remarks to the Author):

I thank the authors for responding in detail to the request to minimize the sequence and structural homology between the training and test sets. Whilst I think their approach goes some way to addressing this, I don't follow why they didn't use the more standard approach of ensuring a minimum sequence / structural identity between the training and test sets. For example in the ThermoMPNN paper a 25% sequence identity cut-off is used to ensure 'none of the proteins in either Megascale or Fireprot test sets have any homologues in either training set.'

We are grateful to the reviewer for their helpful and constructive feedback throughout the entire process, which has contributed to significantly improving our work in previous revisions. We acknowledge the reviewer's continued concern regarding our dataset splitting strategy. While we agree on the importance of adhering to standard practices, in this particular case, we believe our approach is better suited for minimizing structural homology, rather than just sequence homology. Minimizing sequence homology is something that the reviewer asked us to do during the last revision, and we agree with the reviewer that this is the best scientific choice for our situation. ThermoMPNN clustered their larger mega-scale dataset of 298 proteins using the MMseq2's easy-cluster tool with a 25% identity cutoff, resulting in 162 clusters containing 1-27 members each. They assigned all clusters with homology matches to the training set and then randomly split the remaining proteins into training, validation, and test sets with an approximate 80/10/10 ratio. We agree with the reviewer that this approach by the ThermoMPNN team is a valid approach, and is well suited for minimizing sequence homology. To explore a similar approach to our situation, we applied MMseqs2's easy-cluster to our own dataset of 175 proteins, resulting in 107 clusters of 1-14 members each. We then grouped clusters with homology matches into one set and placed singleton clusters into another, yielding two sets of 91 and 84 proteins, respectively (as shown in the figure below).

Encouragingly, both TM-score averages are relatively low (below 0.5, suggesting that proteins are generally not within the same fold), and the inter-set is slightly lower than the intra-set score, indicating modestly reduced structural homology between the sets. However, when compared to the splitting that we performed for the previous revisions after clustering based on pairwise TM-scores, we observed lower inter-set homology and greater separation between sets. Specifically, between our two sets of 35 and 140 proteins, the average intra-set TM-score was 0.49, while the average inter-set TM-score was only 0.29, indicating a greater minimization in structural homology between sets (see SI Figure 13 below from the paper depicting this information).

In summary, while we recognize that the clustering strategy used by the ThermoMPNN team is likely a valid one, we strongly believe that our TM-score-based hierarchical clustering approach is more effective at minimizing structural homology, which is our primary concern, rather than sequence homology. We would thus ask to not have to change the paper. That said, if the editor and/or reviewer strongly prefers that we implement the MMseqs-based method for dataset splitting, we are open to implementing those splits and retraining both our ThermoMPNN and DMS-Fold models.